# Identification of druggable binding sites and small molecules as modulators of TMC1
Pedro De-la-Torre [1,7,8,9] ✉, Claudia Martínez-García [2,9], Paul Gratias [1,9], Matthew Mun [1,3], Paula Santana [4], Nurunisa Akyuz [5], Wendy González [6], Artur A. Indzhykulian [1,3] ✉ & David Ramírez [2] ✉

Our ability to hear and maintain balance relies on the proper functioning of inner ear sensory hair cells, which translate mechanical stimuli into electrical signals via mechano-electrical transducer (MET) channels, composed of TMC1/2 proteins. However, the therapeutic use of ototoxic drugs, such as aminoglycosides and cisplatin, which can enter hair cells through MET channels, often leads to profound auditory and vestibular dysfunction. To date, our understanding of how small-molecule modulators interact with TMCs remains limited, hampering the discovery of novel drugs. Here, we propose a structure-based drug screening approach, integrating 3D-pharmacophore modeling, molecular dynamics simulations of the TMC1 + CIB2 + TMIE complex, and experimental validation. Our pipeline successfully identified three potential drug-binding sites within the TMC1 pore, phospholipids, and key amino acids involved in the binding of several compounds, as well as FDA-approved drugs that reduced dye uptake in cultured cochlear explants. Our pipeline offers a broad application for discovering modulators for mechanosensitive ion channels.

The sensory hair cells of the inner ear function as mechanoreceptors, converting various mechanical stimuli—including sound-induced vibrations, gravitational forces, and linear acceleration—into electrical signals, mediating our senses of hearing and balance. However, deficiencies or malfunctions in hair cells, stemming from genetic mutations[1,2], aging[3], exposure to loud noise[4], or drug-induced ototoxicity, often result in hearing loss. Notably, platinum-containing chemotherapeutic drugs[5,6] and the aminoglycoside (AG) group of antibiotics are known ototoxic agents that cause hearing loss and balance dysfunction[5,7].

Recent studies suggest that non-AG antibiotics may also cause ototoxicity, although clinical reports of such cases are less frequent[8]. AG ototoxicity may occur due to several factors, including the administration of doses exceeding the therapeutic range or the use of enantiomeric mixtures of AG[9]. Several mechanisms have been implicated in AG-induced hearing loss[10–15]. It is widely recognized that AG uptake into hair cells primarily occurs through the mechanoelectrical transduction (MET) channels, a

protein complex formed between the pore-forming TMC1/2 subunits[16–20], and other binding partners such as TMHS, CIB2, and TMIE[21]. Surprisingly, about 9000 dihydrostreptomycin (DHS) molecules per second are predicted to enter into hair cells in a voltage-dependent manner[22], at therapeutic concentrations[17].

MET channels are gated by force transmitted through tip-link filaments[23,24] composed of cadherin-23 (CDH23)[25,26] and protocadherin-15 (PCDH15)[27–31]. The resting tension applied by the tip link to the MET channel increases its open probability at rest, enabling AG uptake[31,32]. Consequently, disruption of tip links by calcium (Ca$^{2+}$) chelators closes the MET channel, preventing AG uptake into hair cells[20]. These findings inspired numerous studies aimed at identifying MET channel blockers for use as otoprotective compounds, preceding the identification of molecules that form the MET channel complex[19,33–38]. The identification of TMC1 as the pore-forming channel subunit has opened up new venues of research for identifying novel pharmacological agents capable of reversibly blocking the MET channel.

[1]Department of Otolaryngology - Head and Neck Surgery, Harvard Medical School and Mass Eye and Ear, Boston, MA, USA. [2]Departamento de Farmacología, Facultad de Ciencias Biológicas, Universidad de Concepción, Concepción, Chile. [3]Speech and Hearing Bioscience & Technology Program, Division of Medical Sciences, Harvard University, Boston, MA, USA. [4]Facultad de Ingeniería, Instituto de Ciencias Aplicadas, Universidad Autónoma de Chile, Santiago, Chile. [5]Department of Neurobiology, Harvard Medical School, Boston, MA, USA. [6]Center for Bioinformatics, Simulations and Modelling (CBSM), University of Talca, Talca, Chile. [7]Present address: Facultad de Ciencias Básicas, Universidad del Atlántico, Barranquilla, Colombia. [8]Present address: Life Sciences Research Center, Universidad Simón Bolívar, Barranquilla, Colombia. [9]These authors contributed equally: Pedro De-la-Torre, Claudia Martínez-García, Paul Gratias. ✉e-mail: pedro_delatorremarquez@meei.harvard.edu; inartur@hms.harvard.edu; dramirezs@udec.cl

Similar to AG treatment, cisplatin causes permanent hearing loss in a significant portion of treated patients due to its gradual accumulation in the cochlea over months to years[39]. Given the impact caused by ototoxic side effects of both AG and cisplatin, the pursuit for identification of novel drugs to prevent the resulting hearing loss is ongoing[40–42]. Currently, there are at least 17 clinical trials evaluating 10 different therapeutics to prevent AG and/ or cisplatin-induced ototoxicity[43]. Recently, the Food and Drug Administration (FDA) approved sodium thiosulfate (STS) to reduce the risk of cisplatin-induced ototoxicity in pediatric patients[44,45], although its mechanism of action is not fully understood[46]. Studies using animal models, however, suggest that STS does not protect hair cells against AG-induced cell death[47], and it remains unclear whether STS interacts with the MET channel[48–52]. This underscores the need for a structure-guided search for novel otoprotectants to mitigate ototoxicity mediated by the MET channel in order to prevent hearing loss in children and adults.

Multiple studies have focused on designing or structurally modifying small molecules to serve as potential otoprotectants against AG-induced hair-cell loss[19,53,54]. However, their interactions within the TMC1 pore remain largely unknown, and whether these compounds share any common TMC1-binding mechanisms is not well understood[17,18,55–59]. A previously reported molecule screen using a chemical library of 10,240 compounds, identified UoS-7692 among several others as a potent MET

blocker[33] (Fig. 1). This compound demonstrated strong otoprotection against AG in zebrafish larval hair cells and in mouse cochlear explants[33].

While experimental screening of relatively small libraries of compounds typically focuses on their in vitro and in vivo evaluation in hair cell-like cell lines, live zebrafish larvae or in mouse cochlear explants, this laborious and a relatively low-throughput approach limits exploration of the broad chemical space. Interestingly, no in silico data evaluating the binding modes of known MET channel blockers within the TMC1 pore cavity has been reported to date. Therefore, conducting an in silico screen to evaluate the binding of potential MET channel blockers to TMC1, followed by more conventional in vitro experiments to confirm the in silico predictions, is an attractive strategy for discovering novel TMC1 modulators.

The pharmacophore concept, defined as the set of structural features recognized at a receptor site or derived from the structure of a ligand, plays a crucial role in determining the bioactivity of a molecule[60,61], and serves as a valuable tool in drug discovery. However, the application of a ligand-based pharmacophore concept for discovering novel TMC1 modulators remains largely unexplored in the context of drug discovery, and the 3D-pharmacophoric and structural factors involved in TMC1-ligand interactions are not well understood. Given the structural diversity of compounds reported as potential MET channel blockers[19,54] (Fig. 1), conducting a comprehensive study to analyze their shared ligand-based pharmacophoric

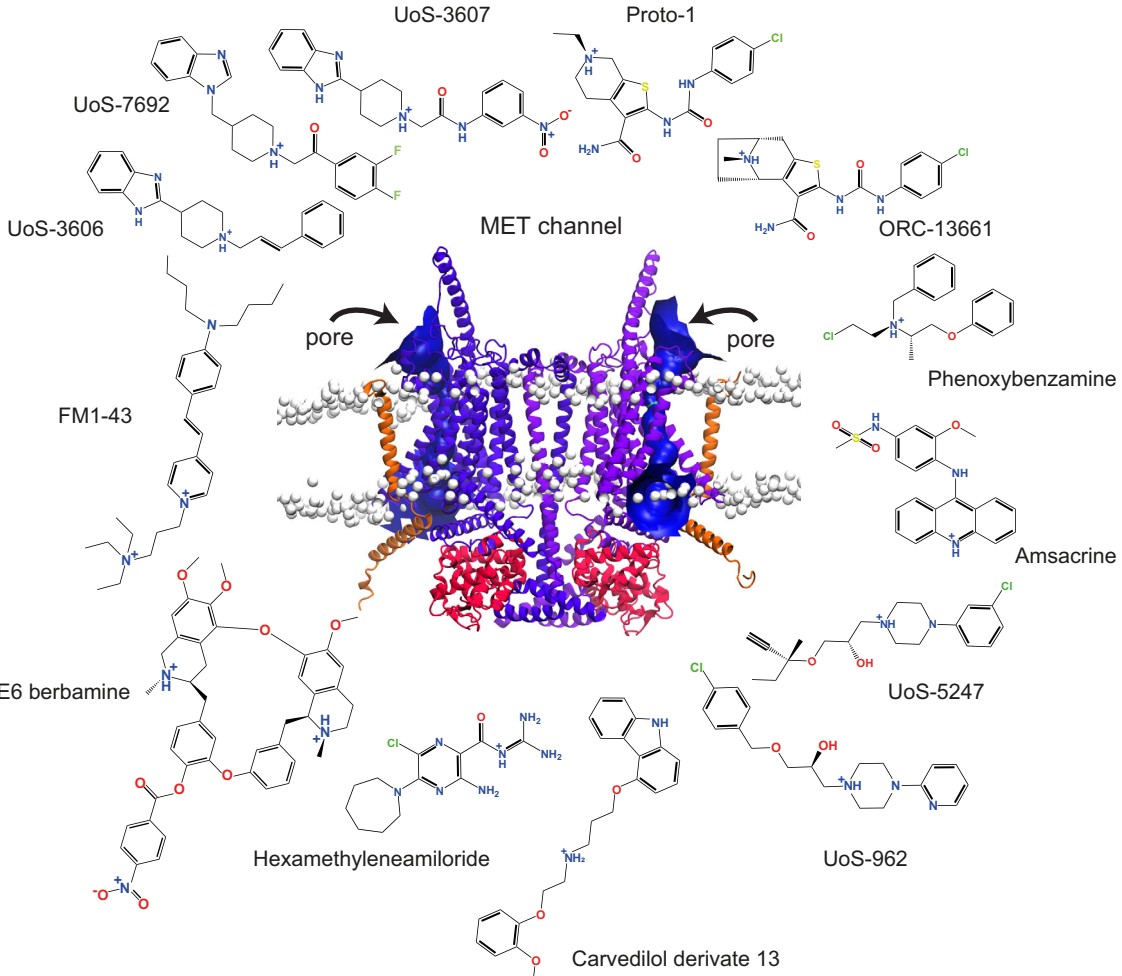

**Fig. 1 | Structural diversity of known MET blockers.** Compounds reported to display varied MET channel blocker potencies and AG protection (see Supplementary Table 1). These compounds share a protonatable nitrogen atom and at least one aromatic ring in their structures. Potential protonatable nitrogen atoms are marked with a *blue* plus (+) sign. Front view of our dimeric TMC1 model (*purple*) in

complex with two protomers of TMIE (*orange*) and two protomers of CIB2 (*red*) proteins. Heads of phospholipids are showed as white beads. Arrows represent the entry site of small molecules via the pores in both TMC1 protomers calculated by *HOLE*[90] (*blue*). More details about this model are presented in Fig. 2 and in the Methods section.

features and their binding to TMC1 is essential for identifying or developing potent and selective MET channel modulators that could potentially serve as otoprotectants. Conversely, considering that TMC1 proteins share structural and sequence similarities with the TMEM16 and OSCA families of membrane proteins[18,55,62–65], potential ligand-based pharmacophores from molecules that modulate their function could also be employed to explore the polypharmacology of these related proteins.

In silico pharmacophore modeling[66–69], molecular docking[70–77], and binding free energy[78–80] studies have emerged as powerful methods for discovering bioactive small molecules. These methods enable a deeper understanding of the key structural factors and moieties required for hit-to-lead optimization with improved biological activities. Previously, we have successfully utilized these methods to design blockers for several ion channels as well as to design other small molecules of biological relevance[81–84].

In this study, we devised a versatile computational strategy to explore the binding modes of known TMC1 blockers and to identify novel modulators of TMC1. Our strategy combines in silico modeling with experimental validation of compounds in mouse cochlear explants. First, we developed common 3D-ligand-based pharmacophore models for small molecules based on known MET channel blockers and predicted their binding modes within the TMC1 druggable pore. These models enabled us to pre-select 258 candidate compounds from over ~22 million compounds (representing ~220 million conformers), sourced from two distinct chemical libraries (non-FDA-approved and FDA-approved drugs).

Using *AlphaFold2*-based structural predictions[72] of the TMC1 protein in an open-like state[85], we conducted a virtual screening (VS) of these 258 molecules, to predict their potential binding modes. Furthermore, we compared 3D pharmacophore features of TMC1 blockers with compounds modulating the activity of the paralog TMEM16A protein, to identify potential structure-pharmacophoric relationships.

We then assessed the binding energies of the docking poses by predicting their binding-free energies using molecular mechanics with generalized Born and surface area (MM-GBSA)[86] methods. From each library, we selected the top 10 hits based on their predicted binding energies and structural diversity for subsequent in vitro evaluation. Next, the MET blocking capacity of each hit candidate was evaluated in vitro using the AM1-43[56] dye uptake assay in mouse cochlear explants using live microscopy imaging. AM1-43, developed as a fixable analogue of FM1-43, functions in live tissue similar to FM1-43, and is often used for both, live and fixed tissue imaging. Our screening pipeline (Supplementary Fig. 1) successfully identified hit compounds that demonstrated a reduction of AM1-43 uptake in cochlear hair cells.

In summary, we developed and experimentally validated a set of effective 3D pharmacophore models and successfully utilized them to identify novel families of MET channel modulators. We then predicted the binding of these compounds within the pore of the TMC1 channel and presented a list of potential binding sites for both known and newly discovered modulators. This structural modeling approach, including molecular dynamics (MD) simulations involving TMC1, CIB2, and TMIE proteins, combined with experimental evaluation in hair cells, allowed us to reveal new putative binding sites critical for ligand interaction within the TMC1 pore. Our in silico approach also predicted flexible properties for TMIE in the MET complex, as well as potential shared pharmacophoric properties between small molecules reported to interact with both TMC1 and TMEM16A proteins[87].

## Results

### Building a good-quality structural model of TMC1 for virtual screening

Homology and experimental structural models for TMC1 and TMC2 have contributed to identifying a putative pore and providing insights into the ion permeation pathway[18,64,85,88]. Cryo-electron microscopy (Cryo-EM) structures of *C. elegans* (*Ce*) TMC1 have confirmed, at atomic resolution, structural similarities between TMCs and TMEM16, as well as the

TMEM63/OSCA protein families[55,62,64,89]. Specifically, TMC1 assembles as a dimer with 10 transmembrane (TM) domains per subunit, similar to TMEM16 proteins[64], while OSCA channels have an additional TM domain, totaling 11 per subunit[65].

In the absence of 3D-atomic structures of the mammalian MET-channel complex, we constructed a model of the complex by assembling *Mus musculus* (*Mm*) TMC1 subunits with *Mm*CIB2 and *Mm*TMIE. *Mm*CIB2 was included to increase the stability of the MET complex and preserve its influence in phospholipid dynamics and ion permeation[21,90]. We utilized the Cryo-EM structure of the expanded *Ce*TMC-1 complex (PDB ID: 7USW)[55] as a reference for structural alignment, and employed the *AlphaFold2* and *Maestro* software[91] for structural modeling. The assembled complex was compared to the *Ce*TMC-1 cryo-EM structure, confirming that the TM10 domain adopted the characteristic domain-swapped conformation[55], which is a structural requirement for the proper oligomerization of *Mm*TMC1 (Figs. 1 and 2). Since some previously reported TMC1 simulations do not include TMIE[21,90], we decided to introduce this component[55] in our modeled mouse MET-channel complex.

Next, the model was subjected to energy minimization, followed by embedding into a pre-equilibrated 1-palmitoyl-2-oleoyl-sn-glycero-3-phosphocholine (POPC) phospholipid membrane and solvation with a final ionic concentration of 0.15 M KCl (Fig. 2) (see *Methods*). Subsequently, the system underwent 25 ns of restrained MD simulation. The Root Mean Square Deviation (RMSD), which measures global structural changes across all backbone atoms, was monitored to assess conformational stability. Equilibration was achieved after 2 ns of MDs, and the RMSD of all backbone atoms remained within a 2 Å distance for the entire complex throughout the remaining simulation time (Supplementary Fig. 2a), indicating a stable conformation of the modeled structure.

When analyzing hetero-subunits independently, no major changes were observed along the 25 ns trajectory, which is in agreement with the literature for MDs of membrane proteins[92–94]. Both TMC1 and CIB2 protomers remained stable during the 25 ns MDs, with RMSD values at ~2 Å. Some fluctuations were also observed in the C-terminal TMIE domain, with RMSD values within 4 Å along the trajectory. This behavior was likely attributable to the "elbow-like" linker[55] as a new flexible component of the MET complex, allowing for the TMIE cytoplasmic helix to move more freely (Fig. 2C and Supplementary Fig. 2a).

Root Mean Square Fluctuation (RMSF) analysis, which assesses local atomic fluctuations and flexibility, revealed that TMC1 chains exhibited fluctuations under 2 Å, similar to its paralog TMEM16A[95] (Supplementary Fig. 2c). Even though TMIE has limited interactions with TMC1, its single-pass N-terminal TM domain fragment presented low fluctuations, likely because it is embedded within the bilayer membrane (Fig. 2C and Supplementary Fig. 2c). In contrast, the C-terminal cytoplasmic helix of TMIE displayed larger fluctuations, likely because it is positioned outside the bilayer membrane and is exposed to solvent, hinging about a flexible region (Fig. 2C and Supplementary Fig. 2c). Similarly, the CIB2 protomers displayed comparable RMSF fluctuations in their N-terminal domains, likely due to the random-coil configuration of these regions (Supplementary Fig. 2c).

The spatial distribution of atoms within the MET complex was assessed using the radius of gyration (*Rg*) measurements (Supplementary Fig. 2b), and no major changes were detected, with *Rg* values remaining within 1 Å, consistent with reports from other 100 ns simulations of membrane proteins. Thus, our results align with previously reported MD simulations of membrane proteins[96]. To further ensure that the stability of the MET complex was indeed achieved within 25 ns, we extended the simulation to 100 ns under identical MD conditions, revealing no major structural changes (Supplementary Fig. 2d–f). This stability can be attributed, in part, to the constraints applied during the simulation, ensuring the stability of the ion channel, and resulting in minimal, if any, conformational differences within the TMC1 protomers and its pores during 25 ns *vs* 100 ns simulation periods (Supplementary Fig. 2g, j). Notably, our *HOLE*[97] analysis demonstrated that the calculated average pore radius (± standard deviation)

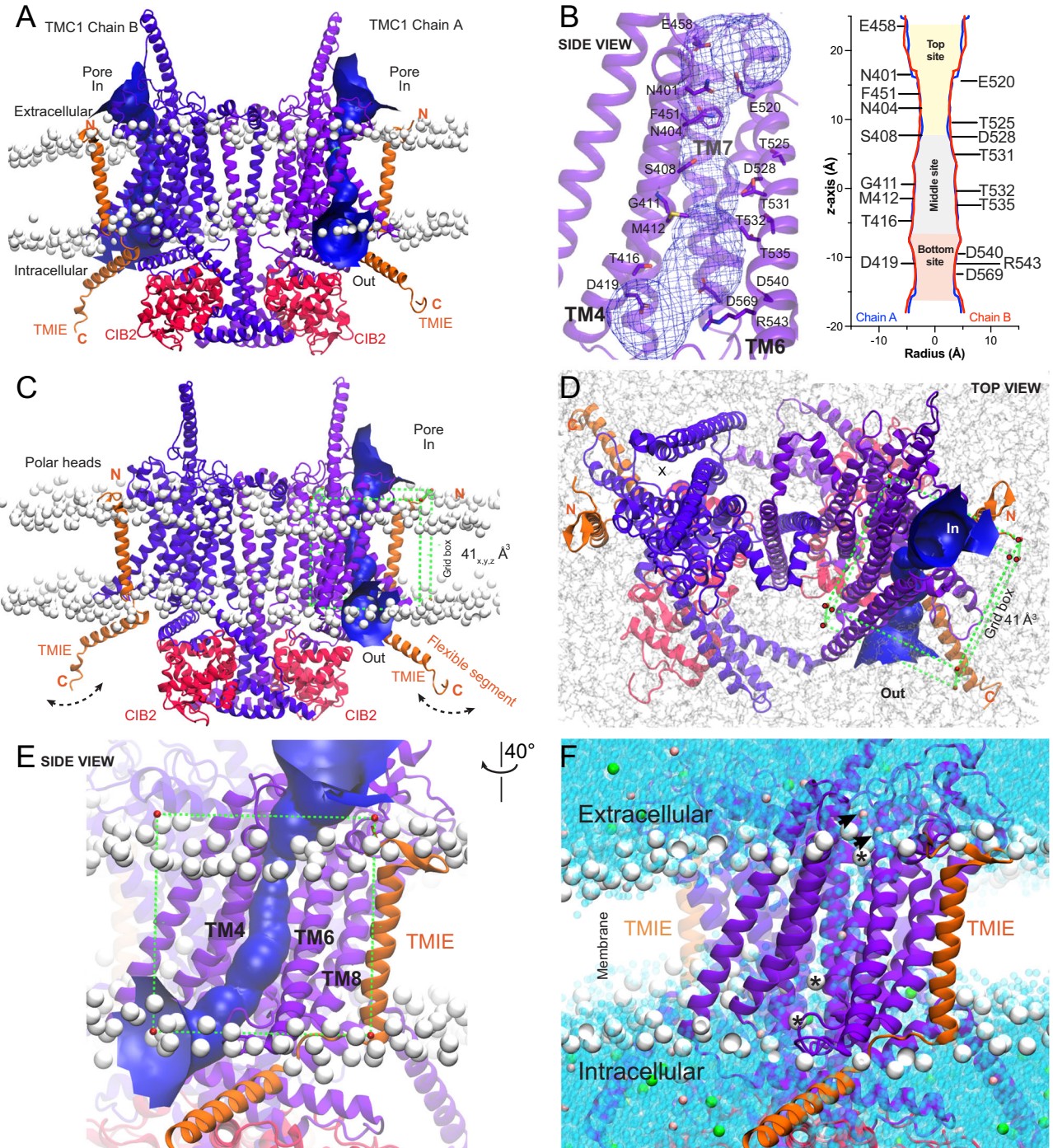

**Fig. 2 | TMC1 modeling and molecular dynamics simulations.** We built a dimeric open-like conformation of the TMC1 structure using *AlphaFold2*. **A** Front view of a 25 ns frame of the dimeric, equilibrated TMC1 protein in complex with two pro-tomers of TMIE (*orange*) and two protomers of CIB2 (*red*) proteins. The system is embedded in a POPC membrane, with the phosphorus of the phospholipids heads illustrated as white beads. The TMC1 pores of chain A and B are represented by a *blue* funnel obtained by *HOLE*[90] analysis. **B** Detailed view of the TMC1 pore of chain A, depicting the pore in a mesh representation, and amino acids as *purple* sticks. The inset depicts the van der Waals radius (Å) of the pore plotted against the distance (Å) along the pore of both TMC1 chains (*z*-axis), obtained by *HOLE*[90] analysis. Top (*gold*), middle (*light gray*), and bottom (*light red*) sites of the pore are labeled and color-coded according to the most expanded regions of the pore. The zero (0 Å) at the *z*-axis represents the reference position of the *middle* site of the TMC1 pore at the

center of the plasma membrane. See Supplementary Fig. 2 for additional details and quantitative analysis. **C** Front view of the complex as in (**A**). The flexible TMIE C-terminal segment is labeled (see Supplementary Fig. 2). A 41 Å³ docking grid box is represented by *green* dashed lines with vertices in *red* circles. **D** Top view of the simulated system showing the swapped TMC1 conformation with two pores represented by a *blue* surface (chain A) and a dashed circle with an X (chain B), respectively. Same grid box as in (**C**). **E** Zoomed-in side view of chain A from **c** showing the grid box and TM domains forming the pore. **F** Zoomed-in side view of the MET complex showing water molecules filling the pore (*blue* beads). K⁺ ions are illustrated as *pink* beads, while Cl⁻ ions as *green* beads. K⁺ ions visiting the pore are pointed with *black* arrows. Phospholipids that moved into the pore are indicated by asterisks (see Supplementary Fig. 5).

**Table. 1 | Pharmacophore models identified by *Phase* and scored by *PhaseHypoScore* (see *Methods*)**

| Pharmacophore ID | Pharmacophore hypothesis | Number of matching compounds | *PhaseHypoScore* |
|---|---|---|---|
| ID-1 | APRR | 7 | 0.780 |
| ID-2 | APR-1 | 9 | 0.762 |
| ID-3 | APR-2 | 8 | 0.759 |
| ID-4 | PRR | 7 | 0.698 |
| ID-5 | ARR-1 | 7 | 0.629 |
| ID-6 | ARR-2 | 9 | 0.526 |
| ID-7 | AHR | 7 | 0.475 |
| ID-8 | ARR-3 | 7 | 0.418 |
| ID-9 | ARR-4 | 8 | 0.385 |
| ID-10 | HRR | 9 | 0.340 |

*A*, hydrogen-bond acceptor, *P* positively charged group, *R* aromatic ring, *H* hydrophobic group.

remained consistent, with no major changes in the pore size distribution. This was assessed at 1 ns intervals for the pore of chain A and compared across the 25 ns and 100 ns trajectories (Supplementary Fig. 2h).

During the 25 ns MD simulations, various phospholipids were observed migrating into the pore, positioning their polar heads near the *top* and the *bottom* of the putative TMC1 pore cavity, with their hydrophobic tails directed towards the pore (Figs. 2F, and 4C, G–I). These lipid movements along the trajectory agree with the findings reported in the literature[55,90], where phospholipids were suggested to move dynamically between the TM4 and TM6 domains of TMC1[21,90], providing further insights into the structural dynamics and lipid-protein interactions within the TMC1 pore.

Furthermore, our analysis revealed the presence of several water molecules and potassium ($K^+$) ions within the pore cavity during the MD simulation, indicating a hydrated pore (Fig. 2F). These findings are in agreement with previous reports, which emphasize the importance of a hydrated environment in facilitating potential drug binding within the TMC1 pore[18,21,55,57,88,90], further supporting our structural model.

The combination of MD simulations and channel pore analysis by *HOLE*[97], has facilitated the identification of three primary target regions within the two elongated TMC1 pores (Fig. 2). Along the *z*-axis, we have assigned these regions to an expanded *top* site located near the extracellular region, a narrowed *middle* site within the transmembrane segment, and a more expanded *bottom* site near the intracellular region (Fig. 2B). The target regions we identified are suitable for screening of druggable-binding sites within the pore and can facilitate a comprehensive conformational search of both known and novel TMC1 interacting molecules using molecular docking (Figs. 1 and 2B–E).

## Identification of common pharmacophores for MET channel block

In this study, we analyzed the chemical structures of known MET channel blockers and identified 3D-pharmacophoric features that contribute to their antagonistic activity (Fig. 1, Table 1, and Supplementary Table 1). These 3D elements include specific functional groups, spatial arrangements, and physicochemical properties that are essential for binding to the MET channel and modulation of its function. The 3D-pharmacophoric models were then used to screen large chemical databases to identify new candidate compounds with similar pharmacophoric features. Suitable hit compounds were then tested in vitro.

We used the Pharmacophore Alignment and Scoring Engine (*Phase*)[68] software, an intuitive pharmacophore modeling tool (see *Methods*) to design 3D-pharmacophore models representing the main structural features of known MET channel blockers. These 3D-pharmacophore features were extracted by assembling a training set consisting of 13 structurally diverse compounds, including UoS-7692[33], UoS-3607[33], UoS-3606[33], UoS-5247[33], UoS-962[33], Proto-1[98], E6-berbamine[54], hexamethyleneamiloride[37], amsacrine[37], phenoxybenzamine[37,99], carvedilol-derivate 13[53], ORC-13661[34,36], and FM1-43[58,59], a positively charged styryl dye often used to label hair cells and to check MET-channel function[58,59] (Fig. 1 and Supplementary Table 1).

These compounds coalesced into 10 predicted pharmacophores (Table 1 and Fig. 3), which were ranked based on their *PhaseHypoScore* values[68]. The top-scoring pharmacophore, designated as APRR (ID #1), achieved a *PhaseHypoScore* of 0.780. High *PhaseHypoScore* values indicate how well the pharmacophoric-feature vectors align with the structures of the compounds in the 3D-pharmacophore model (see *Methods*). All 10 3D-pharmacophores along with their matching compounds, are reported in Supplementary Table 2 and Supplementary Fig. 3.

The APRR pharmacophore consists of four key features: one hydrogen bond acceptor group (*A*), one positively charged group (*P*), and two aromatic rings (*R*) (Fig. 3A). Notably, this model matched 7 out of the 13 known MET blockers reported in the literature, which we used as a training set for this study (Table 1, Fig. 3, and Supplementary Table 2). The remaining nine pharmacophores (ID #2 to #10) were modeled with three pharmacophoric features, reaching *PhaseHypoScore* values between 0.340 and 0.762 (Table 1).

Most of these pharmacophore models shared at least one common aromatic ring (*R*), and one acceptor group (*A*), except for the PRR (ID #4) and HPP (ID #10) models. Furthermore, the top four pharmacophores: APRR, APR-1, APR-3, and PRR (IDs #1 to #4) contained one protonatable group (*P*), matching 7 (APRR), 9 (APR-1), 8 (APR-3), and 7 (PRR) of the 13 known MET blockers, respectively. This suggests that a protonatable amine (positively charged $N^+$ group) is likely crucial for blocking TMC1 activity, consistent with experiments reported in the literature[33,53,57].

To validate our pharmacophore models, we tested their ability to discriminate between decoys and known MET channel blockers. We evaluated the performance of the models using the area under the curve (AUC) of the corresponding receiver operating characteristic (ROC) curve. The AUC value ranges from 0 to 1, with 1 indicating ideal performance and 0.5 indicating random behavior. All AUC values for our models ranged from 0.86 to 0.99 (Supplementary Table 3), demonstrating that our pharmacophore models can accurately classify compounds as active or inactive. Additionally, all the active compounds (known MET-channel blockers) were successfully identified by each pharmacophore model.

We further evaluated the performance of each pharmacophore using the Güner-Henry (*GH*) scoring method. This metric is a reliable indicator, because it incorporates both the percentage ratio of active compounds in the hit list and the percentage yield of active compounds in a database. The number of active and decoy compounds for each pharmacophore, along with the characteristics for *GH* (eg, sensitivity (*Se*), specificity (*Sp*), enrichment factor (*EF*), percentage yield of active compounds (*Ya*), and % yield of actives (*% Yield*) are listed in Supplementary Table 3. Eight out of the ten pharmacophore models have a *GH* score higher than 0.7 indicating that these models are good and reliable[100–103]. Some studies consider a *GH* score greater than 0.5 to indicate a good model reliability[104], suggesting that pharmacophores ARR-1 (GH score = 0.675) and HRR (GH score = 0.664) are also valid models.

## Pharmacophore-based virtual screening of hit compounds

Next, we employed Pharmacophore-Based Virtual Screening (*PBVS*), a computational method for screening large chemical databases to identify molecules that possess pharmacophoric features similar to a reference pharmacophore model[68]. In this study, we screened for compounds that matched any of the 10 pharmacophore models we developed. We used two libraries of compounds as databases: Library 1, which contains over 230 million commercially available, non-FDA-approved compounds (ZINC20 database)[105], and Library 2, which consists of 1789 FDA-approved drugs (MicroSource Discovery Systems)[106].

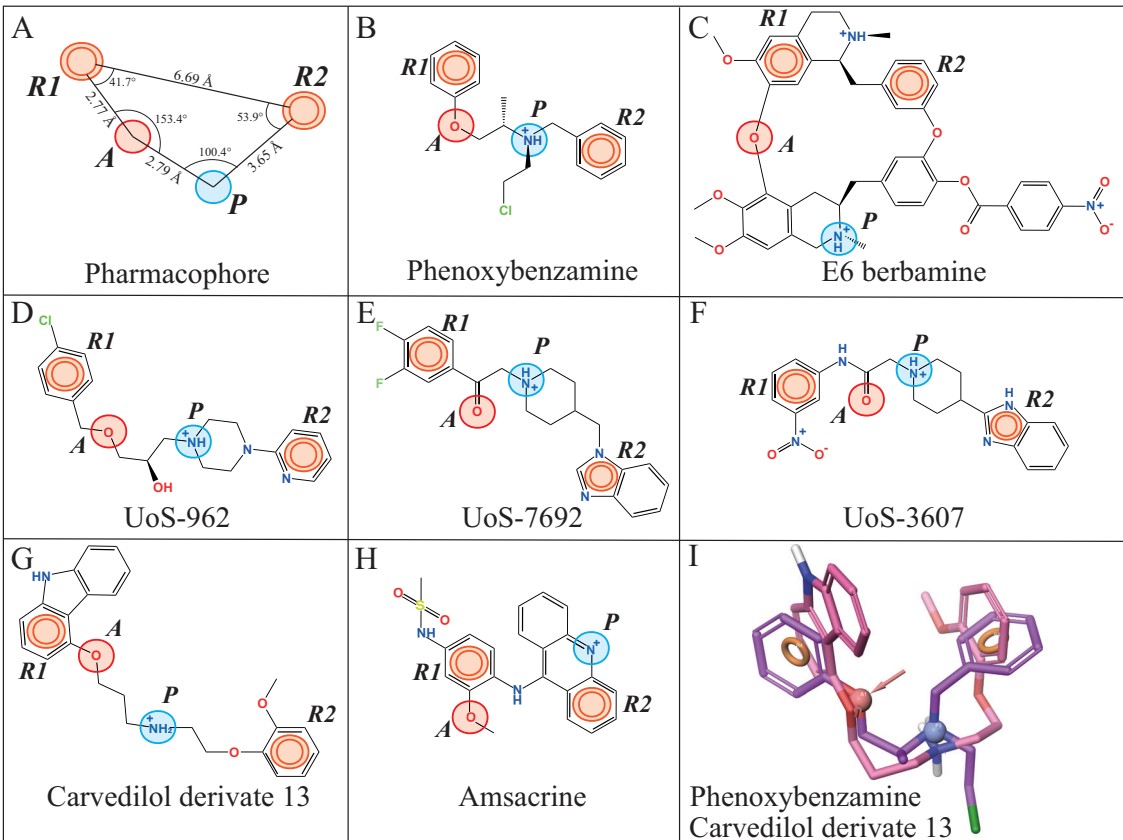

**Fig. 3 | Pharmacophore modeling of 13 known MET channel blockers. A** The APRR four-point pharmacophore model of MET channel modulators show a hydrogen-bond acceptor feature (*A*) in *red*, two aromatic rings (*R1* and *R2*) in *orange*, and a positively charged group (*P*) in *blue*. Distances and angles between the pharmacophoric features are labeled. This model showed the highest *PhaseHypo-Score* (APRR = 0.780) matching 7 out of 13 matching blockers. **B–H** A 2D-

representation of the APRR pharmacophore model with matching compounds. **I** Superposed 3D-structures of the MET blockers phenoxybenzamine (*purple*) and carvedilol derivative 13 (*pink*) onto the APRR pharmacophore. These two compounds fit all 10 pharmacophores reported in Table 1 (see also Supplementary Table 2).

*PBVS* was carried out using two consecutive steps. In the first step, we used *ZINCPharmer*[107], a free pharmacophore search software, to screen commercial compounds from the ZINC database[105]. The second step involved the *Phase* (see *Methods*), which employs a *PhaseScreenScore* function that evaluates both the quantity (partial compound matching) and quality of ligand feature matching. This evaluation includes site matching, pharmacophore feature alignment, and volume-scoring components for each compound[108].

The first step of our *PBVS* analysis involved inputting the 10 pharmacophore models into the *ZINCPharmer*[107] (see Table 1 and Supplementary Fig. 3). We then selected the top 5000 compounds with the lowest RMSD, indicating a good fit to the respective pharmacophore model, resulting in a total of 50,000 pre-filtered compounds (5000 compounds × 10 pharmacophore models). After pharmacophore matching with *ZINCPharmer*, we selected only those compounds that matched with two or more pharmacophore models to increase the reliability of our *PBVS*. This filtering resulted in 4187 compounds from the ZINC database that matched with one or more pharmacophore models (Supplementary Table 4).

Next, the 4187 chemical structures were processed using the *LigPrep*[109] and *Epik*[110,111] modules of the *Maestro* software[91] (See *Methods*) to structurally optimize the dataset and perform the second step of *PBVS* with *Phase*. *LigPrep* module was used to generate 3D structures and optimize their geometry, while the *Epik* module predicted up to 10 different optimized structures (i.e. conformations) per each of the 4187 compounds, accounting for protonation and ionization states, tautomers, and chiralities at specific pH levels (see *Methods*). This increased the total number to 4501

processed molecules, including 314 additional conformational pairs, identified with different protonation or structural conformations.

Subsequently, we performed the second step of *Phase-PBVS* screening and calculated the highest *PhaseScreenScore* values from the 4501 processed molecules corresponding to each of the 10 pharmacophore models. Of the 314 additional molecules that represented various confirmations of the original compounds, we selected the best *PhaseScreenScore* value across all conformations for each compound, reducing the library down to 4187 optimized molecules (Supplementary Table 4).

The results were ranked based on their *Total-PhaseScreenScore* ($TPS_S$) ranging in values between 1.157 and 21.512. These values were then used to calculate the docking threshold ($D_T$), set to $TPS_S$ score values above the mean plus two standard deviations (Mean + 2 SD), as a requirement for selecting compounds for final molecular docking (see *Methods*, Eq. 1). Of the 4187 optimized compounds from Library 1, many aligned with multiple pharmacophore models (Supplementary Fig. 4a): 1625 compounds matched six of the ten different pharmacophores, while 16 optimized compounds matched only two pharmacophores. After applying the $D_T$ threshold, 207 effective compounds were selected for molecular docking to predict their binding modes (Supplementary Fig. 4b).

A similar second *Phase-PBVS* screen was performed for compounds from Library 2 using *Phase*. First, the 1789 FDA-approved drugs underwent ligand optimization (including the assignment of charges, determination of protonation states, tautomer, and chirality determination) using *LigPrep*[109] and *Epik*[110,111] modules of the *Maestro* software[91] (see *Methods*). This resulted in a total of 2628 optimized drug molecules, including 839 additional conformations, reflecting different protonation states, conformers,

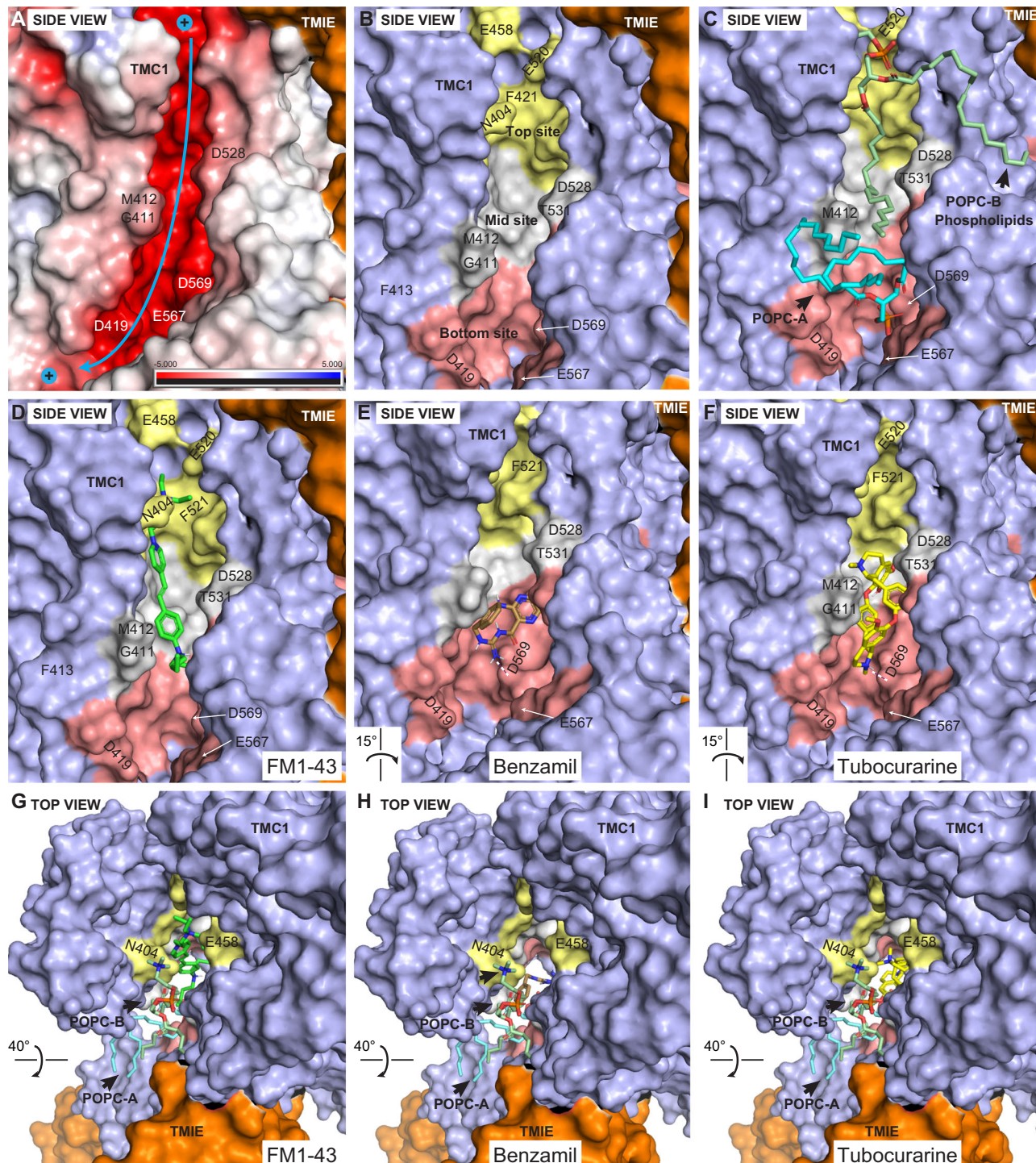

**Fig. 4 | Molecular docking of known blockers within the TMC1 pore cavity.** Side view surface representation of TMC1 chain A pore (*light purple*). TMIE is shown as an *orange* surface, while CIB2 is not displayed for clarity. **A** Surface electrostatic potential (calculated by *PyMOL*) of TMC1 cavity. Electronegative area is highlighted in *red*, electropositive in *blue*, and hydrophobic in *white*. The cavity of TMC1 is highly electronegative with a cation pathway indicated by the *blue arrow*. **B** TMC1 pore cavity highlighting 3 ligand-binding sites: *top* (*gold*), *middle* (*light gray*), and *bottom* (*light red*) sites of the pore, color-coded for illustration purposes. Binding at these sites is governed by interactions between the positively charged amine group of the ligands with residues E458 and F451 (*top site*), D528 (*middle site*), and D569 (*bottom site*), as well as hydrophobic and hydrogen bond interactions with other residues within the TMC1 pore cavity. **C** Side view, similar to (**B**), showing the location of two key phospholipids (POPC-A and POPC-B) identified during MD

simulations. The polar head of POPC-A points towards the site forming a zwitterionic-like interaction network near D569, key for ligand binding. Additionally, the polar head of POPC-B forms a similar zwitterionic-like network with R523 at the *top* site (see also Fig. 6 and Supplementary Fig. 5). **D** Binding interaction of FM1-43 (*green*) across the *top*, *middle*, and near the *bottom* sites of the pore. **E** Binding interaction of benzamil (*brown*) at the *bottom* site of the pore. Dashed *white* line represents a hydrogen bond interaction with D569. **F** Binding interaction of tubocurarine (*yellow*) at the *middle* and the *bottom* sites of the pore. Dashed *white* line as in (**E**). **G–I** Top views of FM1-43, benzamil, and tubocurarine showing the location of the ligands and the phospholipid sidewall formed by POPC-A and POPC-B surrounding the ligands. Additional illustrations on interactions are presented in Fig. 6 and Supplementary Fig. 5.

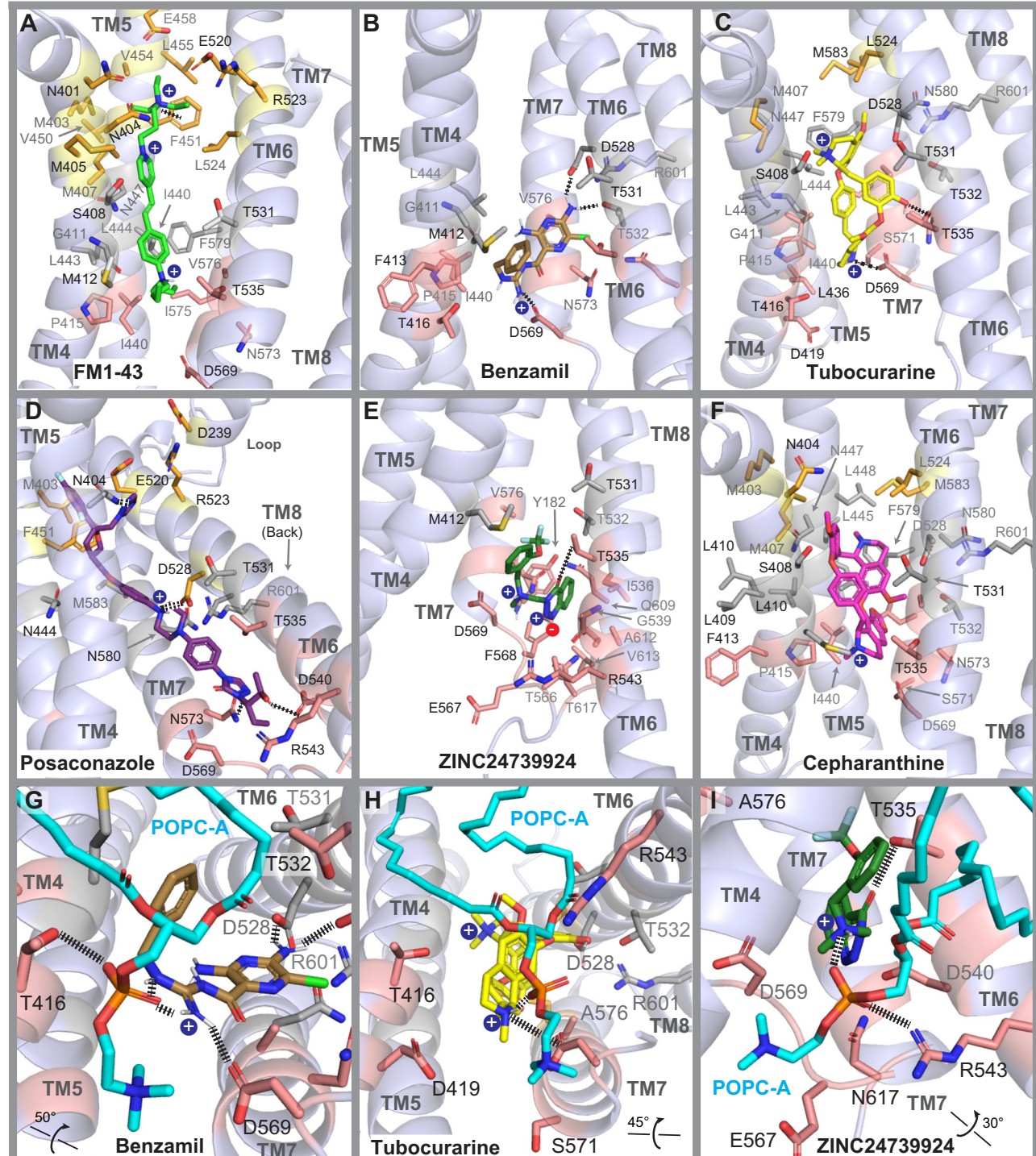

**Fig. 5 | Molecular docking simulations of known and novel TMC1 blockers.** Cartoon representation and binding modes of blockers. **A** FM1-43 (*green*) within the cavity of TMC1 (*light purple*). **B** Benzamil (*brown*). **C** Tubocurarine (*yellow*). **D** Posaconazole (*dark purple*). **E** Compound ZINC24739924 (*green*). **F** Cepharanthine (*magenta*). **G–I** Bottom view of the ZN3 *bottom* site with benzamil, tubocurarine, and compound ZINC24739924, respectively. Residues are labeled in black and gray for their position at the front or at the back of each helix, respectively. The side chains and the backbone residues are colored as in Fig. 4, according to their location within the *top* (*gold*), *middle* (*light gray*), and *bottom* (*light red*) sites of the TMC1 pore. Residues within 5 Å distance from each blocker are displayed, and key interactions highlighted with *black* dashed lines. Additional illustrations on interactions are presented in Supplementary Fig. 5.

stereoisomers, and enantiomers for each compound. All 2628 optimized drug molecules were used for *PBVS* to determine their *PhaseScreenScore* and *TPS_S* values. For the 839 additional molecules, the best *PhaseScreenScore* value was selected, reducing the library down to the 1,789 structurally optimized drug molecules (Supplementary Table 5).

Among the 1789 structurally optimized drugs, 906 matched to one or more pharmacophore models: 332 matched one of the ten different pharmacophores, 574 aligned with two or more pharmacophores, and 26 of those optimized drugs matched all ten pharmacophores (Supplementary Fig. 4c, and Supplementary Table 5). Conversely, the remaining 883

drugs were excluded by the *PBVS* as they did not match any pharmacophores. The *TPS_S* values for the matching drugs ranged between 0.419 and 18.593 (Supplementary Table 5). Subsequently, we calculated the $D_T$ values to select compounds that will pass for the next step of molecular docking, identifying 53 effective compounds (Supplementary Fig. 4d). In this second step of *PBVS*, the optimized molecules from Library 1 exhibited higher *PhaseScreenScore* results compared to those from Library 2 (Supplementary Fig. 4). This is likely because, unlike Library 1, Library 2 is smaller and was not pre-screened using *ZINCPharmer* in the first step of *PBVS*. In contrast to the ZINC database, the smaller size of the FDA-approved dataset made it computationally feasible to screen all compounds directly.

A closer examination of the structural features of each pharmacophore model and the matching selected compounds from Libraries 1 and 2 suggests that the best-scoring ligands adhere to three or four common point-pharmacophoric features. In Library 1, the APRR model was the highest-ranked pharmacophore model for the compound ZINC26876007, with a *PhaseScreenScore* of 2.449 (Supplementary Table 4). In Library 2, carvedilol was the highest-ranked compound, scoring best with both, the APR-1 model (*PhaseScreenScore* = 2.301) and the APRR model (*PhaseScreenScore* = 1.736) (Supplementary Table 5). The identification of carvedilol as a potential compound further validates our *PBVS* design, as carvedilol was previously reported as a MET channel blocker[37,53], despite not being included in our training set of compounds used to design the pharmacophores (Supplementary Tables 1 and 2).

## Predicting binding sites of known MET blockers within the pore region of TMC1

Next, we used molecular docking and molecular mechanics with MM-GBSA methods to predict the binding sites and affinities of 16 known MET blockers within the pore region of TMC1 (see *Methods*).

This set included 13 blockers used in the training set for building 3D-pharmacophore models (Fig. 1, Supplementary Table 1), as well as three known potent MET blockers: benzamil[33,38,112–114], tubocurarine[19,114], and DHS[17,22,115,116]. These three blockers were not part of the training set due to their structural similarities with other blockers, such as functional moieties shared with E6-berbamine and hexamethyleneamiloride (Fig. 1), despite their high potency as MET channel inhibitors.

To explore the conformational space of the TMC1 pore, we performed docking simulations using our modeled "open-like" *Mm*TMC1 structure in complex with CIB2 and TMIE. We used HOLE analysis to identify and characterize the pore, allowing us to precisely target the transmembrane region for docking simulations (Fig. 2). A grid box of $41_x \times 41_y \times 41_z$ Å³ was positioned to cover the pore region. The placement of the grid box is based on solid experimental evidence indicating that the pore of TMC1 is a pathway for both ion and small molecule uptake into hair cells. Thus, the grid box effectively encompassed the chemical space of the TMC1 pore, making it suitable for molecular docking. Docking simulations were carried out with the membrane bilayer from equilibrated 25 ns MDs. During the docking process, ligands were allowed to move freely without constraints (Fig. 2).

Two phospholipid molecules (referred to as POPC-A and POPC-B) were identified near the *top* (residues R523 and N404) and the *bottom* (residues T416, D419, R543, and D569) sites of the pore. These phospholipids were observed to enter the pore cavity and form a sidewall between TM4 and TM6 along the pore region (Figs. 2F, 4C, and 5g–i), potentially modulating the accessibility and binding of small molecules with their polar and hydrophobic moieties.

Using the standard precision (*SP*) scoring function to enrich the conformational sampling of the compounds within the TMC1 pore, we predicted a total of 178 docking poses for the 16 MET blockers mentioned above. Each blocker generated up to 10 poses, including the *R* and *S* stereoisomers for tubocurarine and phenoxybenzamine. These poses were evaluated using the *Emodel* score[117] (see *Methods*), which reflects the quality of each cluster of poses.

All compounds successfully docked within three main areas of the pore, which we have named the *top*, *middle*, and *bottom* sites (Figs. 2B and 4). At the *top* site, key residues involved in binding include F451, E520, R523, S527, and the phospholipid POPC-B. At the *middle* site, important residues include M407, S408, M412, N447, D528, T531, T532, and R601. At the *bottom* site, key residues identified are T535, D569, and the phospholipid POPC-A. A detailed list of key binding residues and their interactions with the blockers, including benzamil, tubocurarine, and DHS are presented in Supplementary Table 6.

**FM1-43 predictions.** FM1-43, a positively-charged fluorescent dye commonly used to test for functional MET channels[57–59], displayed interactions across all three regions of the TMC1 cavity, particularly with residues of the TM4 and TM5 helices (Fig. 5A). At the *top* site of the pore (depicted in *gold*), FM1-43 is positioned between F451 and N404 and exhibits a 4.14 Å cation–π interaction between its positive triethylammonium group and F451 (Fig. 4D and 5A). This group is also surrounded by the negatively charged residues E458 and E520, engaging in a zwitterionic-hydrogen bond network (referred to as the ZN1 site) formed by D239, R523 and the polar head of POPC-B at the entrance of the pore (Fig. 5A and Supplementary Fig. 5). Additionally, hydrophobic interactions with M403 and M407 further stabilize FM1-43 in this region.

Within the *middle* site of the pore (Fig. 4, shown in *light gray*), the hydrophobic aromatic core of FM1-43 interacts with the tail of POPC-B, which further stabilizes the ligand within the cavity. The benzene group of FM1-43 forms close hydrophobic contacts with residues G411 (TM4), L444 (TM5), as well as V574 and L575 (TM7). At the bottom of the pore (Fig. 4, shown in *light red*), the positively charged dibutylamine group is positioned near the polar head of POPC-A, pointing towards residue D569. Since both the docking and MM-GBSA methods use a dielectric field to implicitly simulate water molecules, we acknowledge the possibility that water molecules may participate in the binding interactions of the dibutylamine group with D569 and POPC-A if simulated with explicit solvent (Supplementary Fig. 5).

Notably, docking results predicted that the positively charged atoms of FM1-43 are located inside the pore, while some structural moieties are docked outside the *HOLE*-pore contour, closer to the phospholipid wall. This suggests that ions and ligands may follow distinct permeation pathways while sharing common key amino acid residues involved in both cation and ligand binding (Supplementary Fig. 6).

**Tubocurarine predictions.** The alkaloid tubocurarine exhibited interactions at the *middle* and *bottom* sites of the pore (Fig. 4F, I). Like FM1-43, tubocurarine displayed analogous interactions around the TM4, TM6, and TM7 domains of TMC1. The positive dimethyl-ammonium group of the tetrahydroisoquinoline moiety is positioned near residues S408 and N447, and a hydrophobic interaction was observed with residue F579 (TM7) within the *middle* site of the pore (Fig. 5C). The phenolic group linking the two tetrahydroisoquinoline scaffolds forms a hydrogen bond with residue T535 and is in close contact (4.12 Å) with T531 on TM6. T535 is adjacent to a second identified zwitterionic-interaction zone, composed by D528, T531, T532, R601, and N580, which we refer to as the ZN2 site (Supplementary Fig. 5).

In the *bottom* site, the protonatable amine of the second methyl-tetrahydroisoquinoline scaffold forms a dual salt bridge with D569 at a distance of 4.22 Å, and a hydrogen bond with the phosphate group of POPC-A at 2.82 Å, a component of a third zwitterionic-interaction network (ZN3) (Fig. 5H). The distances were measured between heavy atoms and the polar $N^+$ group.

**Benzamil predictions.** Benzamil primarily engages in significant interactions at the *bottom* site (depicted in *light red*) of the pore cavity. The amine group at position 3 of the pyrazine ring forms dual hydrogen bonds with D528 and T532 on TM6. These residues, along with R601 (TM8), N598, and N580 (TM7), constitute the ZN2 site, which effectively

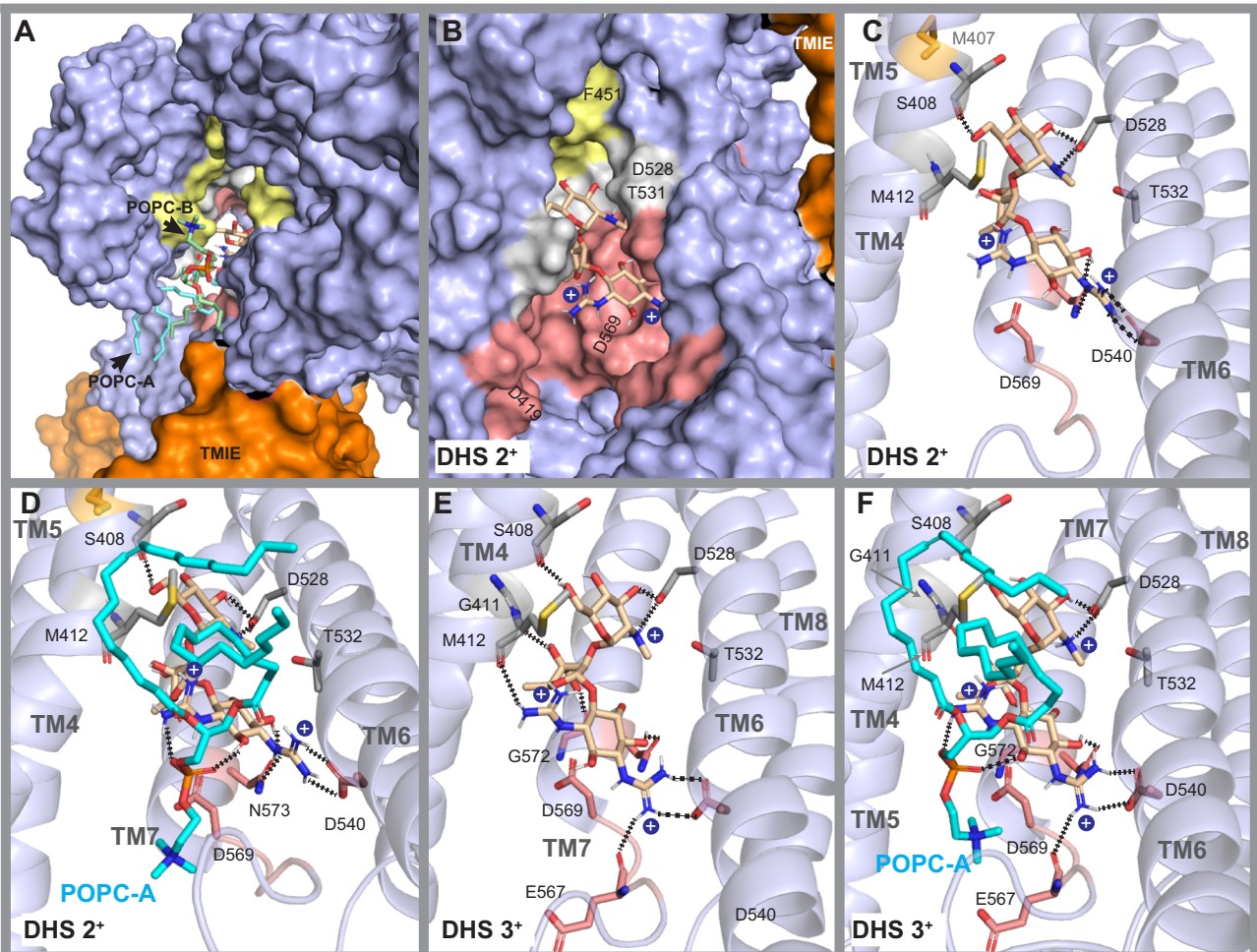

**Fig. 6 | Molecular docking of dihydrostreptomycin (DHS) within the TMC1 pore.**
Binding interactions following molecular docking and MM-GBSA. Phospholipids
are not displayed in some panels to visualize DHS and TMC1 only. Each positively
charged nitrogen is shown as a *blue* bead. Dashed lines represent direct hydrogen
bonds or salt-bridge interactions. **A** Top view of DHS ($2^+$) showing the location of
the ligand and the phospholipid sidewall formed by POPC-A and POPC-B sur-
rounding the ligand. **B** Side view of DHS ($2^+$) within the TMC1 pore, showing the
*top* site (*gold*), *middle* site (*light gray*), and *bottom* (*light red*) sites. Phospholipids
are not displayed for clarity. Binding at these sites is governed by interactions between
the positively charged amine and guanidinium groups, as well as the hydroxyl
groups of DHS ($2^+$) with amino acids in the *middle* and *bottom* sites. **C** Docking pose of
DHA ($2^+$) showing hydrogen bonds between the *N*-methyl-*L*-glucosamine head and
the amino acids S408 and D528. The streptose moiety points towards TM7, while the

guanidinium groups of the streptidine moiety displayed hydrogen bond interactions
with N573 and salt bridges with D540. One of the guanidinium groups points to
D569 in a solvent-exposed region (*more details in* (**D**)). **D** Bottom view from (**C**),
showing interactions between DHS ($2^+$), TMC1, and POPC-A within the *bottom* site
of the pore cavity. One guanidinium group displayed interactions with N573 and
D540 (as in (**C**)), while the second guanidinium group showed interactions with the
polar head of POPC-A. **E** Side view of DHS ($3^+$) within the pore showing interactions
similar to DHS ($2^+$). Additionally, the hydroxyl group of the streptose moiety
formed hydrogen bonds with the carbonyl backbone groups of G411 and G572,
while the guanidinium groups formed hydrogen bonds with the carbonyl backbone
groups of M412 and E567. **F** Interactions of DHS ($3^+$) within the pore and with
POPC-A.

clamps and stabilizes the pocket formed between TM6, TM7, and
TM8 (Fig. 5B and Supplementary Fig. 5). The 6-chloropyrazine ring is
located near residue N573 (TM7) and Q609 (TM8). The
N-benzylcarbamimidoyl group interacts with residue M412, the site of
the M412K Beethoven mutation on TM4[118], as well as with the hydro-
phobic tail of POPC-A, forming van der Waals interactions.

In addition, the positively charged amidino group forms a salt bridge
with D569 (2.82 Å) and a hydrogen bond with the phosphate head of
POPC-A (3.1 Å). This phosphate head further establishes a hydrogen bond
with T416, thereby expanding the ZN3 interactions at the *bottom* site
(Fig. 5G and Supplementary Fig. 5). Our results indicate that the ZN1, ZN2,
and ZN3 interaction zones play crucial roles in ligand binding and
TMC1 stabilization at the *top* (along with F451), *middle*, and *bottom* sites of
the TMC1 pore, respectively.

Notably, these zwitterionic zones contain known residues implicated in
ion permeation, such as D528 and D569, which are essential for ligand

binding, blocking the TMC1 pore, as well as TMC1 protein
expression[85,119,120]. In addition, our results suggest that residues T532 and
T535 are essential for hydrogen bonding interactions with benzamil and
tubocurarine, respectively.

**DHS predictions.** *LigPrep* and *Epik* predicted two main charged forms of
DHS, consistent with structures reported in the literature[121,122]. One form,
referred to as DHS $2^+$, carries two positive charges due to the protonation
of the two guanidinium groups in the streptidine moiety (Fig. 6A–D). The
second form, DHS $3^+$, has an additional positive charge from the pro-
tonated *N*-methyl-*L*-glucosamine moiety (Fig. 6E, F).

DHS $2^+$ displayed interactions towards the *middle* and *bottom* sites of
the TMC1 pore (Fig. 6B, C). Like most of the middle-site interactions
analyzed above, the neutral *N*-methyl-*L*-glucosamine moiety of DHS $2^+$
formed double hydrogen bonds with D528 on TM6, and the backbone
carbonyl of S408 on TM4 at the middle site. Within the *bottom site*, the

**Table 2 | Re-scoring of the docking energies for the selected hits from *Library* 1 and *Library* 2 ($\Delta G_{bind}$ = kcal × mol$^{-1}$)**

| *Library* 1 (non-FDA-approved compounds) | | | *Library* 2 (FDA-approved drugs) | | |
|---|---|---|---|---|---|
| Compound | Without phospholipids $\Delta G_{bind}$ | With phospholipids $\Delta G_{bind}$ | Compound | Without phospholipids $\Delta G_{bind}$ | With phospholipids $\Delta G_{bind}$ |
| **ZINC58438263** | −52.49 | −70.02 | **Posaconazole** | −59.84 | −104.39 |
| **ZINC12986242** | −46.71 | −59.54 | **Cepharanthine** | −53.79 | −40.41 |
| **ZINC06530230** | −44.86 | −50.16 | **Indinavir** | −53.76 | −58.04 |
| ZINC07001403 | −42.20 | −70.48 | **Nefazodone** | −46.31 | −66.65 |
| ZINC12756822 | −40.36 | −67.09 | **Lapatinib** | −45.73 | −19.26 |
| **ZINC12430014** | −38.22 | −77.04 | **Suvorexant** | −39.55 | −39.03 |
| ZINC33126270 | −35.82 | −47.73 | **Pantoprazole** | −30.72 | −42.62 |
| ZINC12890205 | −32.04 | −36.62 | **Pyritinol** | −25.42 | −37.32 |
| ZINC64590918 | −28.47 | −51.42 | **Amitraz** | −24.10 | −30.02 |
| **ZINC24739924** | −22.42 | −39.93 | **Ceforanide** | −20.68 | −23.29 |

Compounds in bold were evaluated experimentally, upon commercial availability.

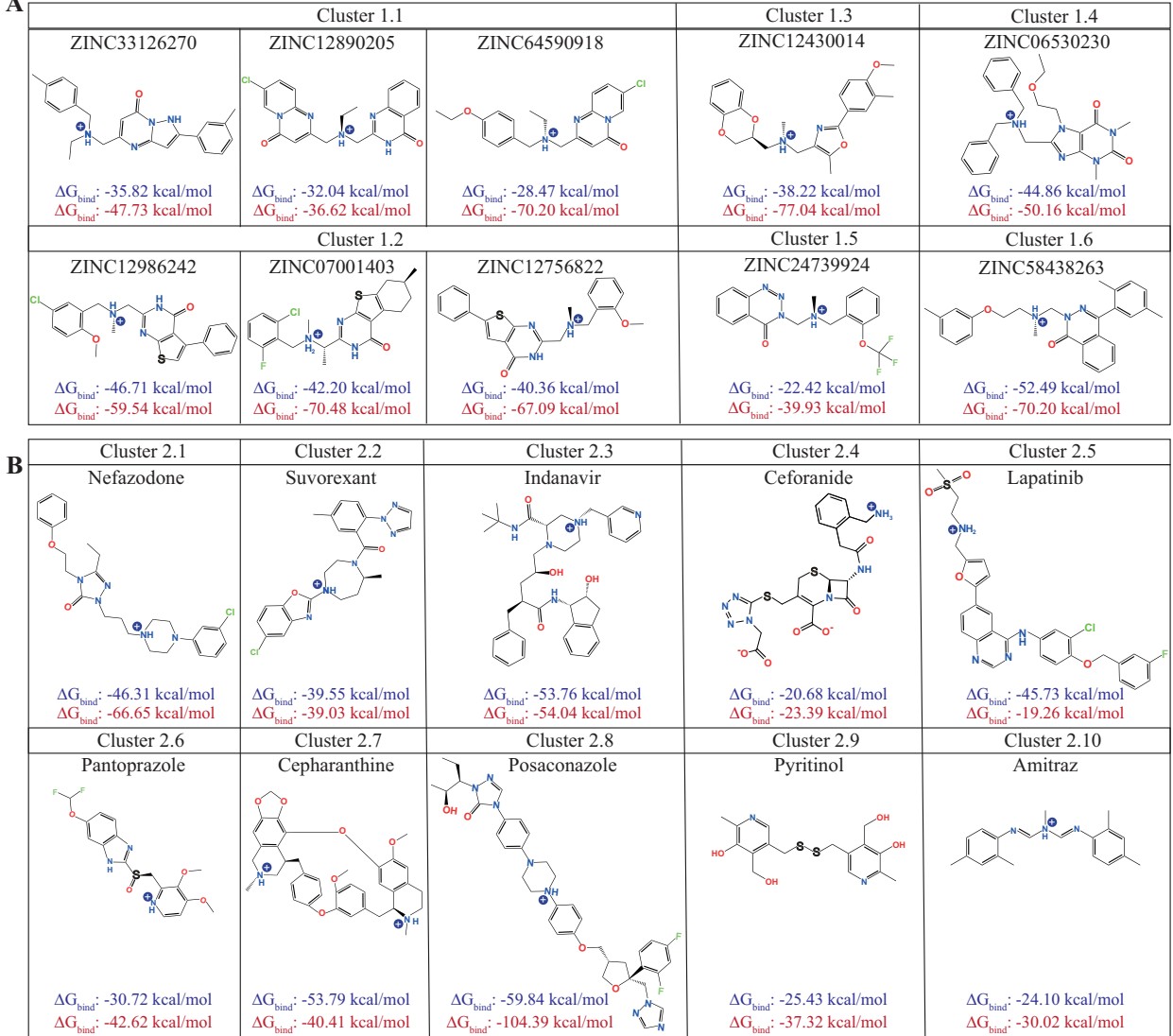

**Fig. 7 | List of final hits selected for experimental evaluation. A** Chemical structures of the 10 selected hits from *Library* 1 (non-FDA-approved). **B** Chemical structures of the 10 selected hits from *Library* 2 (FDA-approved drugs). The calculated MM-GBSA $\Delta G_{bind}$ energies are shown for all selected hit compounds in *blue* (without POPCs) and in *red* (with POPCs), respectively. Compounds were sketched using *ChemDraw*.

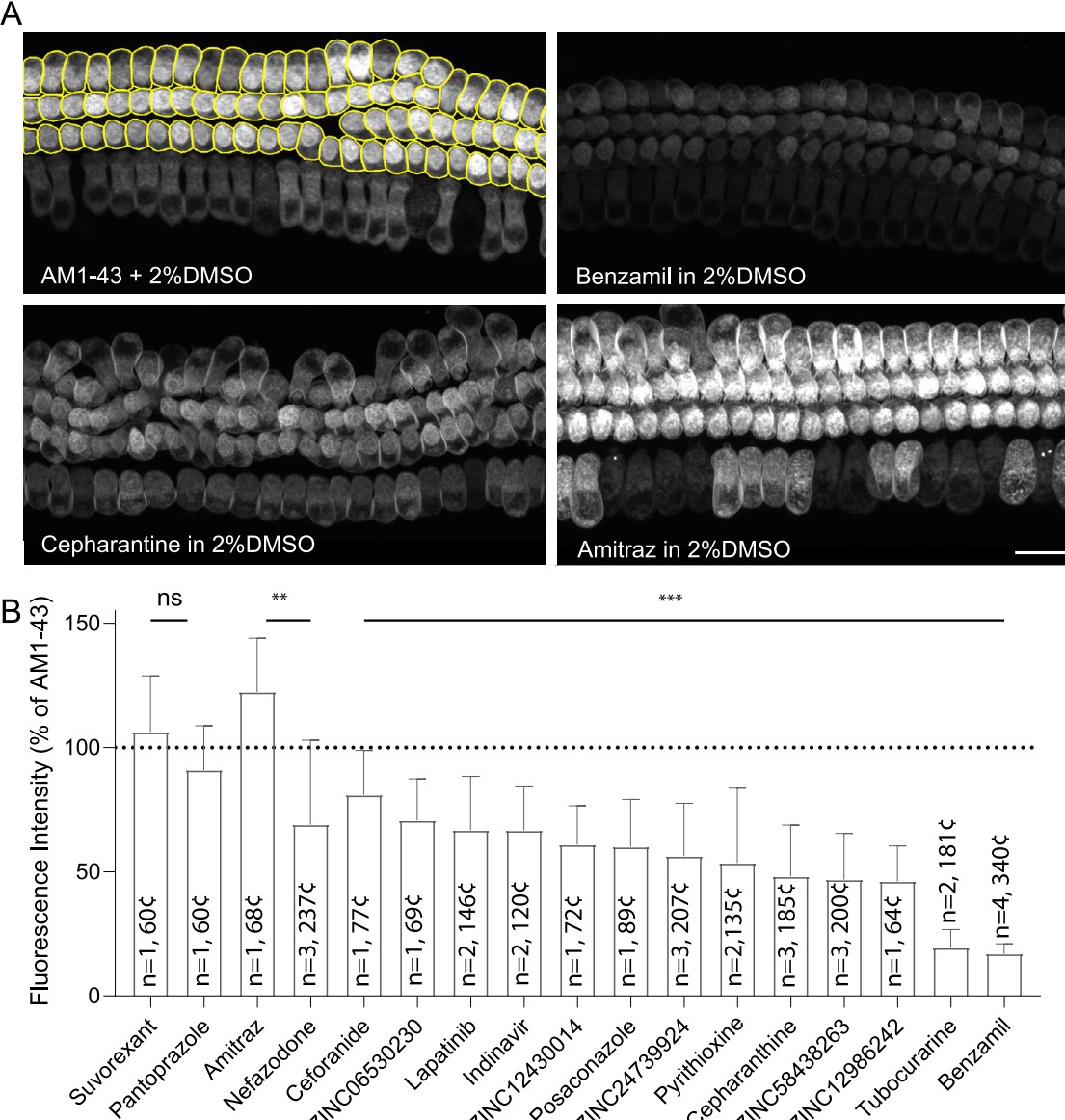

**Fig. 8 | Evaluation of predicted TMC1 modulators in live cochlear hair cells using AM1-43 dye uptake assay. A** Representative confocal microscopy images of AM1-43 dye loading into hair cells from the *middle* region of the mouse cochlea. The *Cellpose* algorithm was used to segment each OHC as individual region of interest (*top left panel*). Dye uptake was reduced by treatment with benzamil (*top right*) and cepharantine (*bottom left*) but increased following treatment with Amitraz (*bottom right*) compared to DMSO-treated controls (*top left*). Scale bar: 15 μm. **B** Averaged AM1-43 dye uptake by OHCs in cochlear explants treated with 15 commercially available hit compounds normalized to values obtained from DMSO-treated control

explants of the same experimental session. The number of experimental sessions and total number of analyzed cells are indicated within the histogram bar. Data are presented as mean ± SD. To compare the average compound fluorescence with the average control levels from the same experimental sessions, the Kruskal-Wallis test followed by Dunn's multiple comparison test was used. $p > 0.5$ (ns); $p < 0.01$ (**); $p < 0.001$ (***). Individual AM1–43 dye loading intensities across each experimental sessions are reported in Supplementary Fig. 8.

hydroxyl and the guanidinium groups of the streptidine moiety show direct interactions with N573 on TM7, D540 on TM6, and the negatively charged head of POPC-A (Fig. 6D).

Similarly, DHS $3^+$ displayed direct interactions with residues D528, S408, D540, N573, and the phospholipid head of POPC-A. Additionally, the streptose moiety formed hydrogen bonds with the backbone carbonyl groups of G411 on TM4 and G572 on TM7. One guanidinium group of the streptidine pointed towards T416, forming direct interactions with the backbone carbonyl of M412 on TM4, while the second guanidinium group interacted with the backbone carbonyl of E567 (Fig. 6E, F).

Our findings propose DHS-TMC1 interactions through residues G411, G572, S408, D540, N573, and E567. These newly predicted inter-acting residues, along with the well-characterized M412, T416, and D528

interaction sites of DHS[88,115], further validate our docking predictions and provide strong confidence in the predicted docking poses of other compounds studied in this work. Furthermore, our predictions also suggest that phospholipids may play a key role in DHS binding. Overall, our results for DHS interaction within the TMC1 pore are consistent with previously reported results obtained using in vitro electrophysiology data[88,115], thereby reinforcing the validity of our predictions (Fig. 6).

**Phospholipid effect on ligand binding affinities in the TMC1 pore: MM-GBSA rescoring analysis**

During the MD simulations, phospholipids (specifically POPC-A and POPC-B) were found to form a lipidic "sidewall" between two transmembrane domains, TM4 and TM6. This close proximity to the pore raised the

question of whether the presence of these phospholipids influences the binding affinity of blockers to the TMC1 protein.

To investigate this, we first re-scored the docking results using MM-GBSA to predict the binding strength of the 16 MET blockers to the TMC1 protein in the absence of phospholipids. We calculated the relative binding free energies ($\Delta G_{bind}$) for the ligands (the known 16 blockers, listed in Supplementary Table 7) also using MM-GBSA. We then investigated whether interactions with POPC-A and POPC-B affected the TMC1-ligand binding energies (Supplementary Table 7). $\Delta G_{bind}$ calculations performed both with and without phospholipids indicated that the presence of POPC-A and POPC-B generally leads to more thermodynamically favorable $\Delta G_{bind}$ energy values for most of the compounds tested.

Among the compounds, FM1-43[57–59] (Figs. 4D and 5A) and E6-berbamine[54], which have distinct structural scaffolds yet fitting similar pharmacophores (Figs. 1, 3, Table 1, and Supplementary Table 2) exhibited the strongest binding affinities with and without phospholipids (Supplementary Tables 1 and 7), consistent with their reported channel blocking potencies[54,57–59]. While hexamethyleneamiloride showed a slightly higher $\Delta G_{bind}$ value, UoS-7692 displayed a less negative $\Delta G_{bind}$ value, possibly due to the presence of the two fluorine substituents in acetophenone moiety at *meta* and *para* positions. Overall, our molecular docking and MM-GBSA analyses suggest that phospholipids may enhance the strength of TMC1-blocker interactions.

## Binding modes and affinities of newly identified hit compounds within the TMC1 pore

We next implemented the same in silico strategy to the compounds obtained from PBVS in the two libraries.

*Library 1 (non-FDA-approved compounds)*: our initial analysis focused on the best 200 compounds from Library 1 with the lowest MM-GBSA $\Delta G_{bind}$ values, without considering phospholipids. However, only 45 of these compounds were commercially available for subsequent in vitro experiments (Supplementary Table 8). Using the Tanimoto-similarity coefficient[123] (see *Methods*) we identified structurally diverse compounds, resulting in 15 different molecular clusters (See cluster IDs in Supplementary Table 8). From these, we selected hit compounds based on binding affinities from clusters that included 3 or more compounds. Specifically, three hits with the lowest $\Delta G_{bind}$ values (without phospholipids) were selected as representatives from clusters 1.1 and 1.2, while one hit was selected to represent clusters 1.3 to 1.6 (Supplementary Table 8).

*Library 2 (FDA-approved drugs)*: Among the 53 compounds analyzed from Library 2, all of which were commercially available, the Tanimoto-structural diversity analysis provided 10 different molecular clusters (see cluster IDs in Supplementary Table 8). One compound with the lowest $\Delta G_{bind}$ value (MM-GBSA without phospholipids) was selected from each cluster for further evaluation.

We subsequently verified whether the selected hits identified through MM-GBSA and Tanimoto analysis without phospholipids also exhibited thermodynamically favorable MM-GBSA $\Delta G_{bind}$ energies in the presence of phospholipids. As with the known blockers (Supplementary Table 7), MM-GBSA results in the presence of phospholipids generally showed improved $\Delta G_{bind}$ energies for hit compounds from both libraries. This confirmed that the selected hits after MM-GBSA and Tanimoto analysis were consistently top ranked in both MM-GBSA subgroups, with and without phospholipids (Table 2). Ultimately, 10 hits were selected from each library for experimental evaluation in cultured cochlear explants (Table 2, Figs. 7 and 8).

Next, we analyzed the binding interactions for selected newly identified compounds within the TMC1 pore, focusing on their binding within the *top*, *middle*, and *bottom* sites of the pore, as guided by *HOLE* analysis and molecular docking (Figs. 2B and 5). Below, we describe interactions for some representative molecules.

Posaconazole is predicted to establish key contacts within the TMC1 pore (Fig. 5D). Its molecular length, measured using the *Maestro* software package, suggested an extensive length of 28 Å (end-to-end distance), allowing posaconazole to interact across the pore. At the *top* site, the triazole

ring forms a hydrogen bond with N404 and polar interactions with the phosphate head of POPC-B at the ZN1 site. In the *middle* site, the dichlorophenyl-furan moiety of posaconazole stacks against M403, M407, and F451. The aromatic rings display van der Waals interactions with residues on TM5, TM6, and TM7 where the positively charged piperazine ring forms a salt bridge with D528 (3.02 Å) in the ZN2 site. At the *bottom* site of the pore, the triazinone ring forms a hydrogen bond with N573 and the hydroxyl group interacts with D540 in the ZN3 pocket.

Another non-FDA-approved compound, ZINC24739924, docked at the *bottom* site of the pore, forming close contacts with residues of TM8 (Fig. 5e). The benzotriazinone moiety of ZINC24739924 stacks in the ZN3 pocket between the polar head of POPC-A, T535, D540, and N573. The charged triazine group makes close contacts with D540, F568, and N573. In addition, the positively charged amine forms a hydrogen bond with the phosphate group of POPC-B, while the benzyl moiety forms van der Waals contacts with M412 and the tails of both POPC-A and POPC-B, stabilizing the ligand (Fig. 5E, I). Furthermore, the trifluoro-methoxy substituent exhibits polar interactions with T535 (3.16 Å) of the ZN3 site. A similar binding pattern was observed for another novel TMC1 modulator, ZINC58438263, which has a 3-methyl-anisole moiety and binds to the ZN3 site of TMC1 (Supplementary Fig. 5).

Finally, the FDA-approved drug cepharanthine shared similar interactions to its homolog alkaloid tubocurarine within the *top*, *middle*, and *bottom* sites of the TMC1 pore (Fig. 5F). Similar to FM1-43, molecular docking and *HOLE* showed that its positively charged nitrogen is located within the predicted pore radius of TMC1 (Supplementary Fig. 6). At the *top* site, the methoxybenzene moiety displays hydrophobic interactions with N404 and polar interactions with S408 (3.98 Å) on TM4. In the *middle* site, the aromatic rings stacked towards the TM4, TM5, and TM7 domains, positioning the methylated methoxy-dihydroisoquinoline and the dioxo-lane moieties between the TM5 and TM7 helices near residues L444 (TM5) and F579 (TM7). The second protonated dihydroisoquinoline is positioned towards the TM4 helix close to M412 (3.44 Å), G411 (3.73 Å), and the hydrophobic tail of POPC-B (3.49 Å). Unlike tubocurarine, cepharanthine did not form a direct interaction between its protonated amine and D569 or the polar head of POPC-A. However, it is possible that water molecules could facilitate these interactions at the cytoplasmic region of the ZN3 pocket (Fig. 5F).

## Identification of key residues modulating TMC1-ligand interactions

We conducted a comprehensive structural analysis of residues within a 5 Å radius of each docked compound within the pore cavity to identify key TMC1-ligand interactions. This analysis included 16 known MET channel blockers, and 20 hit compounds from Libraries 1 and 2 (Supplementary Tables 7, Table 2, and Fig. 7), totaling 36 compounds. We examined all TMC1 residues lining the pore based on their contact frequency with the ligands, as determined by our docking and MM-GBSA pipeline analysis. Residues were categorized as high-contact (contact frequency >0.5) or low-contact (contact frequency <0.5), indicating whether more than 50% or less than 50% of evaluated ligands interacted with each residue, respectively (Supplementary Fig. 7, with residues scoring above or below the *red-dashed line*).

Notably, several previously characterized residues known to influence TMC1 channel function, such as *Mm*TMC1 M412, D528, T531, and D569[18,90,115,119,124–127], were identified as high-contact-frequency residues in this study, suggesting their accessibility for ligand interactions. More importantly, we also identified a set of novel residues with contact frequencies exceeding 0.5, indicating their likely involvement in modulating TMC1-ligand interactions. These residues include M407, S408, G411, P415, T416, I440, L444, N447, L524, T532, T535, G572, N573, A576, F579, M583, and R601.

Additionally, both POPC phospholipids showed the highest contacts frequencies, suggesting that they may play a significant role in modulating TMC1 function. In summary, our pipeline, which combines molecular

docking with MM-GBSA, effectively predicts high-contact-frequency residues that are likely to modulate TMC1-ligand binding across the three druggable binding sites along the TMC1 pore.

## Validation of novel TMC1 modulators in cochlear hair cells using AM1-43 dye loading assay

To validate our in silico findings, we conducted a fluorescent dye loading assay using the FM1-43 analogue, AM1-43, to assess the ability of newly identified compounds to block the MET channel in murine cochlear hair cells. When briefly introduced into the bath solution, these large, positively charged fluorescent dyes enter hair cells through MET channels, which are open at rest. As such, FM1-43 and AM1-43 are commonly used as indicators of MET channel activity, allowing us to evaluate the effectiveness of each compound in interacting with the pore and modulating the dye loading through the MET channel (Fig. 8).

For these experiments, cochlear explants from postnatal day 3 (P3) mice were cultured for two days in vitro at 37 °C and 8% $CO_2$. The explants were then exposed to either AM1-43 alone (positive control, supplemented with 2% DMSO) or in combination with the compound for 60 seconds, following a prior 60-second pre-incubation with the compound. After rinsing off excess dye and neutralizing background fluorescence with the 4-sulfonate calix[8]arene sodium salt (SCAS) quencher, live imaging of the explants was conducted using confocal microscopy.

The fluorescence intensity levels of outer hair cells were individually quantified using the *Cellpose* algorithm[128] and normalized to the average values measured from explants treated with AM1-43 only, separately for each experimental session. As an additional control, we incubated some cochlear explants with 100 μM tubocurarine or benzamil, two well-established potent MET channel blockers known to largely prevent FM1-43 uptake[19,112,114] (see *Methods*). All compounds tested in vitro were applied at a standardized concentration of 100 μM (Fig. 8).

Of the 20 structurally diverse hit compounds (Table 2), representing 10 compounds from each library, only 15 were commercially available and thus tested ex vivo. Most compounds (12 out of 15) showed a statistically significant reduction in the AM1-43 dye loading through the MET channel (Fig. 8 and Supplementary Fig. 8). From Library 1, the compounds ZINC24739924 (56.25 ± 21.29%), ZINC12986242 (46.1 ± 14.28%), ZINC12430014 (60.94 ± 15.64), ZINC58438263 (46.8 ± 18.59%), showed significant reductions of AM1-43 uptake. From Library 2, posaconazole (60.02 ± 19.12%), pyrithioxine (also called pyrithioxin or pyritinol) (53.52 ± 30.18%), and cepharanthine (48.09 ± 20.72%) exhibited the most promising results. Despite being less effective, ZINC06530230, nefazodone, indinavir, lapatinib, and ceforanide still significantly reduced AM1-43 loading. Interestingly, amitraz from Library 2 appeared to have the opposite effect, increasing AM1-43 loading into OHCs (122.4 ± 21.64%; Supplementary Fig. 8).

We selected representative compounds that interact within the *top*, *middle*, and *bottom* sites of the TMC1 pore for illustration (Fig. 5): posaconazole, cepharanthine, and ZINC24739924, three effective blockers that reduced hair cell MET-mediated AM1-43 dye loading by approximately 50% (Fig. 8, and Supplementary Fig. 8). Overall, our ex vivo results align with our in silico predictions, indicating that compounds that were predicted to be thermodynamically favorable to interact within the TMC1 pore showed moderate but significant reductions of AM1-43 loading when tested in cochlear hair cells. Further in vitro evaluation with additional biological replicates and single-cell electrophysiology is needed to carefully assess the potency of each hit compound. Thus, this ex vivo results support the efficacy of our in silico pipeline in identifying novel MET channel modulators within a chemical space of millions of compounds and understanding their potential binding interactions within the TMC1 pore.

## Cepharanthine's dual role in TMC1 and TMEM16A modulation suggests shared pharmacophores

Because of the structural and evolutionary relationship between the TMC and TMEM16 families of proteins[18,62,63,129], the structure and electrophysiological properties of TMEM16 proteins have garnered significant interest in studies involving TMCs. Interestingly, cepharanthine, one of the most potent FDA-approved drugs identified in this study for reducing the AM1-43 loading into cochlear hair cells (Fig. 8), has also been previously reported to inhibit TMEM16A[87,130]. Therefore, we investigated whether inhibitors of both TMC1 and TMEM16A proteins share common pharmacophoric features.

To do this, we used *Phase* to virtually screen 10 known modulators of TMEM16A (MONNA, Ani9, TMinh-23, Zafirlukast, Niclosamide, Evodiamine, Tannic acid, theaflavin, $E_{act}$, and $F_{act}$)[87,130] against the 10 TMC1 pharmacophores (Table 1, and Supplementary Fig. 3). Our results indicate that both cepharanthine and theaflavin share the same APRR pharmacophore (theaflavin matches with 9 pharmacophores). Although theaflavin lacks a protonatable amine, *Epik* predicted a protonated carbonyl group instead (Supplementary Fig. 9a–d). Additionally, the ARR-2 pharmacophore exhibited similar vector features between cepharanthine and Ani9, a known inhibitor of TMEM16A (Ani9 matches with 6 pharmacophores) (Supplementary Fig. 9e–h).

Our predicted docking poses for cepharanthine indicate that it binds primarily at the *middle* and *bottom* sites of the *Mm*TMC1 pore, with some interactions at the *top* site near N404 (Figs. 5F, 6E–H, and Supplementary Fig. 9e–g). However, previous studies have predicted and tested cepharanthine and theaflavin binding to TMEM16A specifically towards the *top* site of the pore[87,130].

To better understand these discrepancies, we performed a comparative structural analysis between our open-like state of the *Mm*TMC1 model and the reported *Mm*TMEM16 structure (PDB code: 5OYB)[131] (Supplementary Fig. 9i–l). As reported in the literature, the *Mm*TMEM16 structure was used to predict the binding mode of cepharanthine and theaflavin only at the *top* site of the pore region[87,130], equivalent to the *top* site within the TMC1 pore (Supplementary Fig. 9i–l). Previously reported docking-guided site-directed mutagenesis experiments and in silico MD simulations have shown that mutations at the upper binding pocket of *Mm*TMEM16A attenuate the ability of cepharanthine and theaflavin to inhibit *Mm*TMEM16A currents[87,132].

The predicted *Mm*TMEM16A upper-binding pocket and the reported mutations can explain the decreased affinity of cepharanthine and theaflavin if they only bind at the extracellular pocket of TMEM16A[132]. However, it is possible that open conformations of the *Mm*TMEM16 pore might expose druggable sites at the *middle* and *bottom* areas of the pore, as we predicted for TMC1 in this study (Figs. 4 and 5, and Supplementary Fig. 5).

However, it is possible that the predicted docking poses for *Mm*TMEM16A at its upper pocket site may have been influenced by a closed conformation of *Mm*TMEM16A and/or restricted by the use of smaller grid boxes[87,132] during docking sampling. Consequently, the conformational space explored in *Mm*TMEM16A might not have sampled potential druggable sites of cepharanthine and theaflavin within the *middle* and *bottom* sites of the pore[87,132], as predicted for TMC1 in our study. This is consistent with recent reports of open conformations of *Mm*TMEM16A and newly identified druggable pocket binding sites across the pore[133,134].

## Discussion

TMC1 is a nonselective cation channel which primarily mediates the influx of $Ca^{2+}$ and $K^+$ into hair cells in response to mechanical stimulation. Unlike other cation channels, TMC1 lacks a conventional cylindrical pore selectivity filter[82], instead featuring a long-curved pore cavity that may be partially exposed to the plasma membrane[21,55,90]. This structure allows TMC1 to be significantly permeable to bulky organic molecules, such as FM1-43[57,58], AGs[17] and other large compounds. In fact, compounds as large as 3 kDa dextrans have been shown to permeate through TMC1[135]. However, the atomic-level interactions of these compounds within the MET channel pore remains poorly understood.

TMC1's permeability to aminoglycosides underscores its relevance in the context of aminoglycoside-induced ototoxicity, a major concern in hearing health. Advancing our understanding of how these compounds

interact within the TMC1 pore could lead to the identification of novel otoprotective solutions to mitigate AG-induced hair cell damage.

The main objective of this study was to identify the common structural features of known MET blockers and use this information to develop a pipeline for discovering novel MET modulators. Additionally, we aimed to explore how these modulators might interact within the TMC1 pore. To achieve this, we employed a comprehensive in silico approach that included 3D-pharmacophore modeling, *AlphaFold2* modelling, molecular docking, Tanimoto similarity analysis, and MM-GBSA $\Delta G_{bind}$ analysis. We screened two compound libraries and identified 10 novel non-FDA-approved compounds and 10 FDA-approved drugs as candidates from a pool of over 230 million compounds. Seven of the 15 experimentally tested compounds showed a significant reduction in AM1-43 dye loading into hair cells (Fig. 8 and Supplementary Fig. 8), validating our approach.

Our in silico findings demonstrate that TMC1 possesses an enlarged cavity with druggable ligand-binding sites, capable of accommodating small molecules[55,136]. We estimate that the pore's narrowest dimension is approximately 4.5 Å diameter, which is larger than previously reported pore sizes (Fig. 2B). This enlarged cavity provides multiple binding sites in the *top*, *middle*, and *bottom* sites of the TMC1 pore, which can accommodate a variety of ligands.

A key outcome of this study is the development of universal 3D-pharmacophore models for MET blockers. These models can be used to screen millions of compounds in silico for further in vitro testing in hair cells. To our knowledge, this is the first in silico pipeline that combines molecular pharmacophore modeling, MD simulations, docking, and MM-GBSA analysis to identify novel MET channel modulators. Furthermore, this study also identified novel atomic details of ligand interactions within the TMC1 pore, revealing a synergistic combination of amino acids that form distinct druggable-binding sites at the *top*, *middle*, and *bottom* sites of the TMC1 pore cavity.

Based on the docking poses of FM1-43 and AM1-43 (Fig. 5A, and Supplementary Fig. 5), we infer that the tight packing of both the triethylammonium and the dibutylamine groups at the *top* binding site of the TMC1 pore, along with the presence phospholipids molecules, may explain why bulkier molecules such as FM3-25 fail to block MET currents[58]. Additionally, this tight packing could contribute to the slower permeation of these bulkier molecules compared to the fast uptake of FM1-43 in hair cells[59].

The results presented in this work strongly support the idea that the ligands bind to key residues within the TMC1 pore, including M412, D528, T531, and D569. These residues have been shown influence both the blocking effects of certain compounds (e.g., DHS, FM1-43) and the ion permeability of TMC1[18,90,115,119,124–127]. Mutations in residues such as D528, D569, and M412 are known to cause deafness and alter mechanotransduction current properties of TMC1[85,118,137]. This indicates that residues involved in modulation of the ion conductance may also play a role in TMC1-ligand interactions. However, our docking predictions suggest that the pathways for ion-conduction and ligand-permeation within the TMC1 pore might be distinct, as chemical moieties of several docked ligands were predicted to bind in regions outside the predicted pore contour identified by *HOLE* analysis (Supplementary Fig. 6).

Our docking results accurately predicted several residues that interact with known MET blockers, which have been well-experimentally characterized in mice through point mutations in TMC1[18,90,115,119,124–127]. These mutations have demonstrated the importance of these residues in permeation and block. Future studies involving site-direct mutagenesis and single-cell electrophysiology will allow to evaluate and further validate the novel contact residues identified in this study.

Our MD simulations, docking predictions, and MM-GBSA energy analyses suggest that phospholipids play a key role in small molecule binding, mediating hydrogen bonds, salt bridges, and hydrophobic interactions within the TMC1 pore (Figs. 4C, G–I, and 5G–I). These findings suggest that proper membrane function around TMC1 promotes stable ligand binding. This is in agreement with experimental observations

showing that blockage of TMC1 with high concentrations of benzamil and tubocurarine triggers membrane scrambling and phosphatidylserine externalization in hair-cell stereocilia bundles[114]. Compared to previous simulations of TMC1 and TMC1 + CIB2, which did not include TMIE as part of the MET complex, our results reveal possible functional implications of the "elbow-like" linker of TMIE as a flexible component of the mammalian MET channel. Notably, flexibility has also been reported for the PCDH15 MAD12 domain, another component of the MET complex, which has been shown to exhibit mechanical weakness under force stimulation[138]. Furthermore, while our results suggest that the C-terminal helix segment of TMIE exhibits flexibility, this observation may complement experimental data indicating its role as a target region for phosphatidylinositol 4,5-bisphosphate (PIP$_2$) binding. This region has been implicated in regulating MET currents, and potentially coupling TMIE with PIP$_2$ to the plasma membrane[139]. However, further studies are necessary to directly determine how flexibility might influence its interaction with PIP2 and its functional implications in the MET complex.

Additionally, the docking and MM-GBSA data show M412 in close contact with phospholipids, also reported as one of high-frequency ligand-binding residues. This may help explain why charged mutations like M412K increase annexin V (AnV) signals, as the p.M412K mutation might attract phospholipid heads, promoting membrane scrambling[114]. Our docking interactions of both benzamil and tubocurarine further support the hypothesis that phospholipids act as ligand-binding modulators (Figs. 4–6, Supplementary Fig. 5, and Table 2).

Since the hit compounds were evaluated in early postnatal cochlear explants (P3 + 2 days in vitro), their blocking potency might have been influenced by the overwhelming presence of TMC2 during that developmental stage in the mouse cochlea, which was not considered in our in silico predictions. TMC2 is transiently expressed in mouse cochlear hair cells during early development and at the onset of the hair-cell mechanotransduction, but is gradually downregulated and is replaced by TMC1 around P10[140,141]. Therefore, selective screening and design of TMC1 blockers may necessitate experimental evaluation in mature cochlear hair cells in the absence of TMC2, or in *Tmc2-ko* mice. However, because the screened compounds were based on pharmacophore models of known MET blockers, they likely display some blocking capability for both TMC1 and TMC2. This hypothesis could be further validated using *Tmc1-ko* and/ or *Tmc2-ko* mice[142].

However, the structural exclusion of TMC2 from our in silico pipeline may promote the selection of hit candidates with possible increased selectivity for TMC1, since the putative pore of TMC2 is distinct from TMC1[143]. The TMC2 pore displays decreased hydrophobicity and a smaller pore radius compared to the homologous *middle* site of TMC1. These characteristics may reduce the affinity of TMC2-ligand interactions[143]. It is well known that most small molecules preferentially bind to hydrophobic pockets[144], highlighting the complexity of TMC1-blocker interactions within the druggable-pore cavity and the high variety of amino acids involved in TMC1-ligand binding. This structural complexity suggests that the binding affinities of compounds with MET-blocker properties may vary depending on TMC isoform and, perhaps, even species.

TMCs belong to a larger superfamily that includes TMEM16 and TMEM63/OSCA proteins[62]. Our study not only sheds light on the structural features required to modulate the uptake capacity of TMC1 proteins but also provides key insights on potential pharmacological relationships between TMC1 and TMEM16A proteins, as they share common 3D pharmacophores for their compound antagonists. This study provides additional structural and functional insights into the role of the TMC1 pore in hair cells[145].

In summary, our combined in silico and in vitro experimental study suggest that a pharmacophore composed by two aromatic groups, one acceptor group, and at least one protonatable amine is required for blocking the TMC1 pore cavity of the mammalian hair-cell MET channel. We developed and provided a proof-of-concept validation of a pipeline for discovering novel MET channel modulators. Our pipeline successfully

identified compound antagonists of the hair-cell MET function able to reduce AM1-43 loading into cochlear hair cells in vitro. Further evaluation of the reported compounds will provide more insights into their modulatory capacities and help identify novel MET blockers, assess their potential otoprotective applications, as well as their possible role in TMEM16A modulation. Additionally, future studies using site-directed mutagenesis will help elucidate the role of high-contact druggable sites in TMC1-ligand binding, their impact on hair-cell mechanotransduction function, and their potential influence on TMC1 ion selectivity and conductance.

## Materials and methods

### AlphaFold2 modeling of the MET complex structure
We generated a dimeric structural model of *Mm*TMC1, spanning residues 81–746, adopting an open-like conformation[85] with domain swapping feature at transmembrane domain 10 (TM10). This model was constructed using *AlphaFold2* through the ColabFold GitHub repository[146]. Subsequently, the MET complex was constructed, also using *AlphaFold2*, by combining two TMC1 subunits, each accompanied with CIB2 (residues 1–187) and TMIE (residues 44–118) in a 1:1:1 ratio[55,147,148].

The resulting MET complex model was then compared with the structure of the expanded *Ce*TMC-1 complex (PDB code: 7USW)[55], showing the typical swapped conformation and location of homolog binding partners of the MET complex.

### Molecular dynamics simulations
The modeled MET complex was prepared using the protein preparation wizard module of the *Maestro* suite[110,111,149]. Amino acid protonation states were assigned at pH 7.4 using PROPKA[150]. Subsequently, the complex was subjected to energy minimization and embedded into a pre-equilibrated POPC bilayer, followed by solvation using the SPC water model. To neutralize the system, ten $K^+$ ions were added, and the final ion concentration was set to 0.15 M KCl. MD simulations were performed with *Desmond*[151] and the OPLS4 force field[152].

The *Desmond* default membrane relaxation protocol was applied before simulations. Then, 25 ns of equilibrium MDs in a $NP\gamma T$ semi-isotropic assembly were performed, applying restrictions to the protein backbone (spring constant 10 kcal $\times$ mol$^{-1}$ $\times$ Å$^{-2}$) with constant surface tension of 0.0 bar $\times$ Å. Temperature and pressure were kept constant at 300 K and 1.01325 bar, respectively, by coupling to a Nose-Hoover Chain thermostat[153] and Martyna-Tobias-Klein barostat[154] with an integration time step of 2 fs. Coulombic interactions were calculated using a cutoff distance of 9 Å. Subsequently, three replicates of 25 ns production MDs were performed using the last frame. Restrictions were applied to the protein secondary structure (spring constant 5 kcal $\times$ mol$^{-1}$ $\times$ Å$^{-2}$) under the same conditions as described above. Replicates' statistical analysis is available at the Ramirez Lab Github repository (see *Data availability section*). We also extended the production 25 ns MD to 100 ns to study the stability of the MET complex. The production 25 ns and 100 ns MDs were analyzed using *Desmond* and custom in-house scripts. Visualization was carried out with *VMD*[155] and *Pymol*[156]. Subsequently, we analyzed the pore with the *HOLE* algorithm. Pore radius measurements of the TMC1 chain A were obtained over 25 ns and 100 ns production trajectories to determine the stability of the pore and the TMC1 dimer. Measurements were obtained every nanosecond throughout the simulations. The 25 ns MDs equilibrated structure was used for further virtual screening. In addition, we analyzed the potential location of phospholipids neighboring the TMC1 pore.

### Ligand-based pharmacophore modeling of known MET blockers
Thirteen MET channel blockers with structural diversity reported in the literature such as UoS-7692[33], UoS-3607[33], UoS-3606[33], UoS-5247[33], UoS-962[33], Proto-1[98], E6-berbamine[54], hexamethyleneamilomeride[37], amsacrine[37], phenoxybenzamine[37,99], carvedilol-derivate 13[53], ORC-13661[34,36], and FM1-43 (a positively charged styryl dye often used to label hair cells)[58,59], were selected to design 3D-pharmacophore models of small molecules using an energy-optimized pharmacophore method.

The pharmacophore design is a versatile approach used to extract common chemical features from a set of small molecules with biological function. First, the structures of the 13 known MET channel blockers were sketched using *Maestro*[91] and then prepared using *LigPrep*[109] (Schrödinger, 2021) with the S-OPLS force.

Salts were removed, no tautomers were generated, and compound chiralities were determined from the reference 3D structure. Subsequently, both pharmacophore and ligand mapping were generated from the 13 MET blockers selected, with the *Phase* module[68]. We used six pharmacophoric features: Hydrogen bond acceptor (*A*), hydrogen bond donor (*D*), hydrophobic group (*H*), negatively charged group (*N*), positively charged group (*P*), and aromatic ring (*R*).

To model the pharmacophores, a minimum of 3 and maximum of 5 pharmacophoric features were set. Also, the percentage of pharmacophore matching threshold (number of known blockers that fits to the modeled pharmacophore) of 50% was selected as a minimal criterion for a representative pharmacophore model of the 13 known MET blockers. All other settings in *Phase* were kept as default. Finally, the best 10 pharmacophore models were ranked using the *PhaseHypoScore*[68]. This score measures how well the pharmacophoric-feature vectors align with the structures of the compounds that contribute to the 3D-pharmacophore model.

To validate our pharmacophore models, we use active compounds (known MET channel blockers) and decoys (40 decoys per active) generated with *LIDeB*[157] (Supplementary Table 3). To evaluate the performance of each pharmacophore, different key measures were considered, including the area under the curve (AUC) of the corresponding receiver operating characteristic (ROC) as described[158]. In addition, we calculated the percentage of actives *(% Yield)*, the percentage yield of active compounds (*Ya*), sensitivity (*Se*), specificity (*Sp*), enrichment factor (*EF*), and the Güner-Henry (*GH*) scoring using the method reported in the literature[102].

### Pharmacophore based virtual screening (PBVS)
In this project, we used two molecule libraries. *Library 1* (non-FDA-approved compounds) from the ZINC database (ZINC20) contains over 230 million commercially available compounds in 3D formats and over 750 million purchasable analogs[105]. *Library 2* (1789 FDA-approved drugs — US Drug Collection, MicroSource Discovery Systems)[106] was in-house processed for *PBVS* using the *LigPrep* and *Epik* modules of *Maestro*[91,109,110] with the OPLS3 force field[159]. All possible ionization states were predicted for each compound at pH 7.4 ± 2.0, and their chiralities were retained.

We carried out a two-step *PBVS* with the non-FDA-approved Library 1. The 1$^{st}$-step *PBVS* using the software *ZINCPharmer*[107], followed by a 2$^{nd}$-step screening with *Phase*. For the *ZINCPharmer* screening, the 10 best pharmacophores generated after the pharmacophore mapping step were used to search the most promising compounds from ZINC20 that fits with any of the 10 pharmacophore models.

Compounds were filtered by selecting a single hit as the maximum limit per conformation for each molecule having a maximum RMSD geometric match of 1 Å against each of the 10 pharmacophore models (compounds with RMSD > 1 Å were discarded). The best 50,000 hits (5000 per each pharmacophore model) with molecular weights between 200 and 700 g/mol were selected from ZINC20. Finally, we selected the compounds that matched two or more pharmacophores using the *KNIME* software[160].

For the 2$^{nd}$-step *PBVS*, the compounds from the 1$^{st}$-step (*Library 1*) were processed with *LigPrep*, and ionization states were generated at pH 7.4 ± 0.5 using *Epik*[110,111]. In parallel, the Library 2 (FDA-approved drugs) were also processed with *LigPrep* and *Epik*. Thus, the prepared molecules from both libraries were screened against each of the 10 pharmacophores using *Phase*, following the methodology reported by Gallego-Yerga et al., 2021[153].

For both libraries, the *PhaseScreenScore* values (which evaluate how well the ligands align to the pharmacophore features of the hypothesis)[161] were obtained for each ligand per screening against each pharmacophore. In addition, we implemented a workflow with *KNIME* analytics platform and built a matrix to calculate the *Total-PhaseScreenScore* (TPS$_S$) for each ligand,

represented by the sum of each *PhaseScreenScore* value. Then, we selected hits for molecular docking if the value was higher than the $D_T$ value, according to the Eq. 1.

$$D_T \geq \overline{X} + 2\delta \tag{1}$$

$D_T$ is the docking threshold, $\overline{X}$ is the average $TPS_S$ of the entire *Library* (1 or 2) solutions; and $\delta$ is the $TPS_S$ standard deviation of the entire *Library* (1 or 2) solutions.

## Molecular docking and free energy binding energy calculations

After *PBVS*, molecules from both libraries were docked with the *Glide* software[117] in the pore of the *Mm*TMC1 model (in complex with CIB2 and TMIE) using the frame structure at 24 ns post-equilibration. The protein complex and all the phospholipids were kept for molecular docking, while waters and ions were removed. A cubic grid box of $(41_x \times 41_y \times 41_z)$ Å$^3$ was centered at methionine M412 (chain A) covering the TMC1 pore including part of the TMIE helical protein embedded in the membrane (Fig. 2).

Molecular docking was performed for the selected hits with the best *PhaseScreenScore* criteria from libraries 1 and 2, overpassing the $D_T$ value, as well as for the 13 known MET blockers (UoS-7692, UoS-3607, UoS-3606, UoS-5247, UoS-962, Proto-1, E6-berbamine, hexamethyleneamiloride, amsacrine, phenoxybenzamine, carvedilol-derivate 13, ORC-13661 and FM1-43), benzamil, tubocurarine, and DHS (known experimental molecules used as MET channel blockers) (Figs. 1, 4, and 6). The *Glide SP* mode and the OPLS_2005 force field were used to explore the positional, conformational, and orientational space of the ligands in the TMC1 pore.

A maximum of 10 docking poses were requested for each ligand and the best pose was determined by selecting the conformer with the lowest *Glide Emodel* score followed by a low *Glide Score* value from superposed-like poses[117]. The *Glide Emodel* was prioritized, since *Glide* uses *Emodel* to pick the best pose of a ligand (conformer selection) and subsequently rank them against one another using *GlideScore*[117]. A manual check of each representative pose was carried out to identify common binding sites for each selected pose from each molecular docking cluster.

Molecular docking solutions from both libraries 1 and 2 were rescored by calculating their binding free energies ($\Delta G_{Bind}$). Two docking post-processing strategies were implemented. **Strategy A**: excluding phospholipids neighboring the pore, and **strategy B**: including key phospholipids neighboring the pore that could influence small molecule binding. $\Delta G_{Bind}$ calculations were carried out using MM-GBSA methods[86] with *Prime*[162], combining molecular mechanics energies and implicit solvation models[163]. The MM-GBSA $\Delta G_{Bind}$ between ligands and the TMC1 channel was calculated with the following equations:

$$\Delta G_{bind} = \Delta H - T\Delta S \approx \Delta E_{MM} + \Delta G_{sol} - T\Delta S, \tag{2}$$

$$\Delta E_{MM} = \Delta E_{internal} + \Delta E_{electrostatic} + \Delta E_{vdw}; \Delta G_{sol} = \Delta G_{PB/GB} + \Delta G_{SA}, \tag{3}$$

where $\Delta E_{MM}$, $\Delta G_{sol}$, and $-T\Delta S$ are the changes in the molecular mechanics energy, solvation-free energy, and conformational entropy upon binding, respectively. $\Delta E_{MM}$ includes $\Delta E_{internal}$ (bond, angle, and dihedral energies), electrostatic, and van der Waals energies. $\Delta G_{sol}$ is the sum of the electrostatic solvation energy, $\Delta G_{PB/GB}$ (polar contribution), and non-electrostatic solvation, $\Delta G_{SA}$ (non-polar contribution). The polar contribution was calculated by using the generalized Born model, while the non-polar energy was calculated by the solvent accessible surface area (SASA). The VSGB 2.0[164] solvation model and OPLS force field were used for these calculations. Residues located within 5 Å from the ligands were included in the flexible region, and the rest of the protein atoms were kept frozen.

## Morgan fingerprint and Tanimoto-similarity coefficient

After virtual screening through both libraries, Morgan fingerprint similarity factors[165] were determined for all molecules with the *RDKit* module of the

*KNIME* analytics platform. This extended-connectivity fingerprint based on Morgan algorithm represent molecules as mathematical objects, which allow us to analyze the structural environment of each atom up to a radius of 2 Å[165]. Using the Morgan fingerprint, we calculated the distance matrix with the Tanimoto-similarity coefficient[123], which allowed us to determine the structural similarity of the compounds on a scale from 0 (non-identical) to 1 (identical).

Then hierarchical clustering was performed with the average linkage method. Clusters were selected using the normalized distance threshold of 0.75 for the non-FDA-approved dataset (Library 1) and 0.85 for the FDA-approved dataset (Library 2) to enhance structural diversity. Representative compounds for populated clusters were selected for further evaluation.

## Mouse cochlear explant cultures

All procedures and protocols were approved by the Institutional Animal Care and Use Committee of Mass Eye and Ear and we have complied with all relevant ethical regulations for animal use. Postnatal day (P) 3 CD-1 mice of either sex were cryo-anesthetized and euthanized by decapitation. Inner ears were harvested and organ of Corti epithelia were acutely dissected in Leibovitz's L-15 (L-15, Gibco #21083027) cell culture medium. Following dissection, explants were affixed to glass-bottom cell culture dishes coated with Geltrex basement membrane matrix (Gibco #A1569601, 100 μL/dish) and cultured in DMEM (Gibco #12430054) supplemented with 3% FBS and 10 mg/L ampicillin for 48 h at 37 °C, 8% CO$_2$.

## Dye loading and fluorescence imaging

Cochlear explants from postnatal day 3 (P3) mice were cultured for two days in vitro at 37 °C and 8% CO$_2$. Following gentle aspiration of the culture medium at room temperature, the explants were rinsed with L-15 and were pre-treated for 60 s with the compound at a final concentration of 100 μM (dissolved in L-15 at 2% DMSO). The solution was then aspirated and replaced with a solution consisting of the loading dye AM1-43 (4 μM) and the hit compound (100 μM) of in L-15 with 2% DMSO. Following another 60 s of incubation, the solution was aspirated, and the excess AM1-43 dye was neutralized using a quenching solution composed of 0.2 mM of SCAS quencher in L-15.

Live imaging of the samples was performed using a Leica SP5 confocal microscope equipped with a 40×, 0.8 NA water-dipping objective lens, zoom was set to 2×, resulting in an effective pixel size 189 nm. The laser power and smart gain settings were kept constant across all experimental conditions. During each experimental session, at least one compound and one control sample were imaged.

## Image analysis

Following image acquisition, maximum intensity *z*-projections were generated using *ImageJ*. Subsequently, Cellpose[128] was used to segment individual outer hair cells (OHC) into regions of interest (ROIs). The mean fluorescence intensity of each OHC was then quantified and normalized to the average fluorescence intensity level of the corresponding control OHCs treated with AM1-43 (4 μM) in L15 and 2% DMSO.

## Statistics and reproducibility

To ensure the reproducibility of results, at least three molecular dynamics simulation replicas were conducted, where each replicate was defined as an identical system initialized with a different random seed. Data are presented as mean ± standard deviation (SD). For statistical comparisons, the Kruskal-Wallis test followed by Dunn's multiple comparison test was used for non-normally distributed data. A *p*-value $< 0.05$ was considered statistically significant. The number of experimental sessions and the total number of analyzed cells are indicated within the corresponding histogram bars.

## Reporting summary

Further information on research design is available in the Nature Portfolio Reporting Summary linked to this article.

## Data availability

The PDB file containing the coordinates of the simulated MET channel complex, and the 3D-pharmacophores used in this study are available for download from the Ramirez Lab Github repository: https://github.com/ramirezlab/Drug-design-targeting-TMC1

## Code availability

The computational workflows employed in this study were executed with standard software packages, with all relevant details provided in the Materials and Methods section. The MET complex structure was modeled using *AlphaFold2*, implemented through the ColabFold GitHub repository. MD simulations were performed and analyzed using the standard builds of *Desmond*, and visualization was carried out with *VMD* and *Pymol*. Ligand-based pharmacophore modeling of known MET blockers was conducted with the *Phase* module of *Schrödinger Maestro* 2021-2. For pharmacophore-based virtual screening, we employed both *ZINCPharmer* and the *Phase* module of *Schrödinger Maestro* 2021-2. Molecular docking was carried out with *Glide*, while $\Delta G_{bind}$ calculations were performed using MM-GBSA methods in *Prime*. Following virtual screening, Morgan fingerprint similarity factors were determined for all molecules using the *RDKit* module within the *KNIME* analytics platform. All the specific parameters and methodologies employed are detailed in the Materials and Methods section.

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

## Acknowledgements
We thank Dr. Angela Ballesteros (NIDCD) for her valuable scientific feedback and critical reading of the manuscript draft, and Carlos Peña-Varas (Universidad de Concepción – Chile) for advice on figures and data visualization. We also thank Dr. Robert Fettiplace, Dr. Anthony Ricci, Dr. Jeffrey Holt, and Dr. Gwenaelle Geleoc for their feedback and suggestions. This work was supported by NIH NIDCD R01DC020190, R01DC021795, and R01DC017166 to A.A.I., the American Hearing Research Association (AHRF) Discovery Grant to N.A., the NIH NIDCD T32 DC000038 Training Grant awarded to G. Géléoc, supporting M.M., as well as Chilean National Research and Development Agency (ANID), Fondecyt 1220656, 1230446, FOVI210027 and FOVI240021 to D.R. The funders had no role in study design, data collection and analysis, decision to publish, or preparation of the manuscript.

## Author contributions
N.A. generated AlphaFold models; P.S. and D.R. performed molecular dynamics simulations to relax the models; D.R., C.M.G., W.G., P.S. and P.D. analyzed the MD simulation data; C.M.G. carried out virtual screening and MM-GBSA calculations, P.D., C.M.G. and D.R. performed virtual screening, MM-GBSA analysis, and the selection of compound candidates; P.G. and M.M. carried out cochlear tissue culturing and dye-loading experiments, with assistance from P.D.; P.G. performed confocal imaging; P.G. and A.A.I. analyzed the imaging data; P.D., C.M.G., P.G., A.A.I. and D.R. prepared the original draft of the manuscript; P.D. and A.A.I. edited the manuscript and incorporated feedback from all authors. D.R., P.D. and A.A.I. conceived the study, supervised the project, and secured funding; P.D., A.A.I., D.R. and P.G. conceptualized the study and developed methodology. All authors contributed to the final version of the manuscript.

## Competing interests
The authors declare no competing interests.
