## [Transparent Peer Review file · Communications Biology]

Identification of Druggable Binding Sites and Small Molecules as Modulators of TMC1

Corresponding Author: Dr Pedro De-la-Torre

Version 0:

Reviewer comments:

Reviewer #1

(Remarks to the Author)

1. Brief Summary of the Manuscript

The research conducted by Pedro De-la-Torre seeks to elucidate the binding sites in TMC1 and to identify novel modulators for this target using both *in silico* and *in vitro* approaches. The *in silico* methodology incorporated structure-based screening techniques, including 3D-pharmacophore modeling, molecular docking, molecular simulations, and binding free energy calculations.

2. Overall Impression of the Work

Notable findings from this study include the identification of three potential drug binding sites within the channel pore, the involvement of phospholipids, and key amino acids in modulator interactions. The pipeline was applied to discover novel TMC1 modulators. These potential compounds were also tested *in vitro*. However, the manuscript requires major essential revisions before publication.

3. Specific comments

Point 1: The molecular dynamics (MD) simulations were conducted for a duration of 25 ns. This simulated time is insufficient to ascertain the stability of the protein. The authors are advised to extend the simulation period to a minimum of 100 ns to provide a more robust confirmation of protein stability.

Point 2: The authors need to supply references supporting the statements made, as well as the criteria for assessing MD results in the Methods section. Specifically, references are required for the following assertions:

Line 161-163: "When analyzing hetero-subunits independently, both TMC1 and CIB2 protomers remained stable during the 25 ns MDs, with RMSD values at ~2Å. However, fluctuations were observed in the C-terminal TMIE domain, with RMSD values within 4Å along the trajectory."

Line 166-167: "Root Mean Square Fluctuation (RMSF) analysis, which assesses local atomic fluctuations and flexibility, revealed that TMC1 chains exhibited fluctuations under 2Å."

Line 173-175: "The distribution of atoms within the MET complex was assessed using the radius of gyration measurements (Supp Figure 1C), and no major changes were detected, with the radius of gyration remaining within 1Å,...".

Point 3: The authors state, "The combination of MD simulations and channel pore analysis by HOLE, has facilitated the identification of three primary target regions within the two elongated TMC1 pores" (Lines 186-187). However, the manuscript lacks a detailed analysis of the channel pore by HOLE. The authors should provide comprehensive details regarding this result.

Point 4: In the 3D-pharmacophore modeling, only the APRR pharmacophore comprises four features, while the remaining nine pharmacophores were modeled with three features. Additionally, The authors selected the top 10 pharmacophore models based on the PhaseHypoScore but did not validate the reliability of these models with external databases. Validation results should be provided.

Point 5: The authors should discuss the structure-activity relationship (SAR) of potential compounds derived from their *in silico* results and the correlation of these results with the pharmacophore model.

Point 7: "The goal of your study is to discover TMC1 modulators". When you conducted the in silico approach, how did you determine whether the potential compounds identified after screening would act as modulators or inhibitors? Please provide more information on this.

In conclusion, the study is well structured which can be published in the journal. However, the authors must revise the manuscript with the above comments. Goodluck!

Reviewer #2

(Remarks to the Author)

The authors present a comprehensive study on identifying potential druggable binding sites and small molecules as modulators of TMC1. However, a few minor issues remain to be addressed.

1. The authors used the AM1-43 dye uptake assay to validate the potential TMC1 modulators in cochlear hair cells. However, this assay has some limitations, such as the low sensitivity and the inability to measure the binding affinity of the compounds. More importantly, whether these compounds have non-specific interactions with TMC1 through aggregation is also a concern.
2. In Figure 3, the chemical structure of Amsacrine is not correctly drawn.
3. In Figure 8, I am unsure whether the protonation states of the compounds Pantoprazole and Pyritinol are correctly assigned.
4. Page 11, lines 521-525. I am confused by the inconsistent experimental results of lapatinib, pantoprazole, and pyritinol compounds.

Reviewer #3

(Remarks to the Author)

The introduction and the majority of the results are really well written. They are logical and easy to follow. In contrast, the latter section seems rushed and requires some edits. From the section on "Validation of novel TMC1 modulators in cochlear hair cells using AM1-43 dye uptake assay" to the end of the discussion there are a number of sections I find confusing and not easy to follow, these are listed below with other queries and typos.

Queries and Typos:

Line 120 "of" not needed

Line 125: "AM1-43, developed 126 as a fixable analogue of FM1-43, functions effectively in live tissue along with FM1-43." I'm not sure what you mean by along. Do you mean its acts the same was as FM1-43 or have you added it with FM1-43? I'm assuming the former.

Line 186-190: "three primary target regions within the two elongated TMC1 pores (Figure 2). Along the Z axis, we have assigned these regions to an expanded top area located near the extracellular region, a narrowed middle area within the transmembrane segment, and a more expanded bottom area near the intracellular region (Figure 2A). – By target regions you mean the top middle and bottom sections of the pore? I think this is best represented by Figure 2B right panel.

Line 244-250: It seems that after optimisation you get an extra 314 or 939 molecules but after looking at the phase screen score these were discarded is that correct? Does this always happen if so why bother identifying the extra if they are only going to be discarded.

Line 295: "were" unnecessary

Figure 5 E not I for Benzamil.

Line 358: Cepharanthine is not one of the compounds listed as one of the 16 being tested in this section. Where has this come from? It needs explaining.

Line 406: You state "ligands" it becomes clear later this is the 16 known MET blockers you used above and not the ones you've predicted but can you make that clear at the beginning.

Line 408: Typo "theses"?

Line 425: Typo space between "Library 1with"

Line 442: "(Table 4)" for some reason not in bold here.

Line 447: "Our results are provided in the same order as presented on Figure 6." - confusing can you just say our results are presented in Figure 6 with the known blockers?

Line 489 - "(M407, S408, G411, P415, T416, I440, L444, N447, L524, T532, T535, G572, N573, A576, F579, M583, and R601) with contact frequencies exceeding 0.5, suggesting they are likely to be among the key residues involved in modulating the TMC1-ligand interaction." – are there any deafness mutations in these residues? I sort of assume not given you say they are novel.

Line 499-500: Where in this experiment has FM1-43 been used? Whether this matters depends on the answer to the question I posed above regarding the nature of AM1-43 compared to FM1-43

Line 505-510: This should probably be in the methods section.

Line 511-515: This is largely repeating the methods.

Line 507: Why have you used 2% DMSO? As this isn't a protection experiment and you've also used 2% DMSO in the control it's probably fine but there is data to suggest that high levels of DMSO can be damaging in these types of assay (doi.org/10.1371/journal.pone.0055359, <https://doi.org/10.1016/j.heares.2007.12.002>). In most of these published assays they use concentrations under 1% or even 0.5% if possible.

Figure 8B: This is quite a confusing and misleading figure. If I understand this correctly each circle represents a cell from one culture and the different colour bars are different independent experiments/cultures. So while there is $n = 3$ for Cepharranthine there is only $n = 1$ for Suvorexant, Pantoprazole, ZINC06530230, ZINC12986242, Amitrazm, Cerforanide, Posaconazole and ZINC12430014. While plotting individual cells keeps the variability for the test compounds averaging the control cells to produce the percentage values loses the variability in the controls which makes this a little misleading. The cells in this case are essentially technical replicates. I'd suggest either you change the analysis to take into account this and therefore keep the variability in both the controls and the experimental (e.g. raw values and controls also plotted as one of the bars on your graph followed by a stats model that takes into account the variability in the cells and the variability between cultures/animals for your analysis) or you average the compounds as well. The latter though will mean you only have an $n = 1$ for some of the compounds. You could increase this with more experiments, or can you prove your point without these compounds?

Line 520: Why do the Library 2 compounds have mean and SD numbers but the Library 1 compounds don't?

Line 522: Lapatinib has a mean of 66.74 with a SD of $\pm 21.65\%$ this is stretching your 60% cut-off threshold.

Line 527-528: I assume the numbers on the end of the sentence are as follows Posaconazole $39.98 \pm 19.12\%$, Cepharranthine $54.63 \pm 21.45\%$ and ZINC24739924 $47.07 \pm 23.98\%$. The reason I ask is because it conflicts with what's written in line 521-522 where Posaconazole is $60.02 \pm 19.12\%$ and Cepharranthine is $48.09 \pm 20.72\%$. Also I don't think 526-529 adds to what's written above.

Line 529: I think you should have tested some compounds you didn't expect to block based on your model to see if they did as predicted not block.

Line 543-549: Other than theaflavin and Ani9 the other TMEM16A modulators showed no similar pharmacophores? If so can you please add a line to this effect for clarity.

Line 592: For readability maybe change to "Seven out of 15 hit compounds experimentally tested showed significant reduction of AM1-43 dye loading into hair cells (Supp. Figure 5)"

Line 594-596: I find this sentence a little clunky can you reword please.

Line 604-607: I find this sentence confusing and it doesn't really follow from the section before can you please reword.

Line 620-624: I find this section confusing, did these point mutations result in deaf mice? What do you mean by site directed mutagenesis in this case are you planning on producing mutant mice with specific mutations? I don't know how this differs from the point mutation mice mentioned. This needs more of an explanation.

Line 653: I don't understand the "Thus" as this is a bit of a jump from the previous section. This seems a bit tacked on as a paragraph.

I find the in silico model very exciting with the potential to reduce the number of animals used in this type of research and also save time and money. I would therefore be interested to know if this model will be available to researchers in the field either as an assessable programme they could run on their compounds or if the authors invite researchers to submit queries to them. Either way it would be nice to have this mentioned in the paper.

Version 1:

Reviewer comments:

Reviewer #1

(Remarks to the Author)

Thank you for addressing my comments and providing detailed responses. Your efforts to clarify and improve these aspects of the manuscript are greatly appreciated.

Point 1: I appreciate your efforts to extend the MD simulations to 100 ns and to demonstrate stability between 25 ns and 100 ns. However, it is necessary to include the results of the 100 ns MD simulations in the manuscript for the following reasons:

1. As you mentioned: "Next, we compared the conformations at MD-25 ns and MD-100 ns and detected no major conformational changes (Figure R1, not included in the manuscript)."

The figure only illustrates the protein structures at selected time frames (0-12-24 ns for MD-25 ns and 0-25-50-100 ns for MD-100 ns). What evaluation metrics led to the conclusion of "no major conformational changes"? Why did you choose the conformations at 0-12-24 ns for MD-25 ns and 0-25-50-100 ns for MD-100 ns?

If the goal is to compare conformations throughout the simulation time, it would be more appropriate to identify time frames showing significant changes and to compare these conformations using specific quantitative evaluation.

2. Regarding the RMSD values, it is evident that after 40 ns, the backbone stability was achieved for most proteins, except for TIMEC(A), which did not stabilize during the 100 ns simulation. Specifically, TIMED(B) and CIB2F(B) exhibited significant fluctuations in the initial 40 ns, ranging from 1 Å to 2.5 Å and 2.0 Å, respectively, before stabilizing at these values afterward. These observations indicate that stability of the backbone was generally reached after 40 ns, not 25 ns. Moreover, TIMEC(A) displayed persistent fluctuations and failed to stabilize throughout the entire simulation.

These findings highlight the importance of extending the simulation time to 100 ns and necessitate including this result in the manuscript to provide a comprehensive analysis.

The other points are satisfactory.

Reviewer #2

(Remarks to the Author)

The authors have addressed all my questions. The manuscript in its current form is ready for publication.

Reviewer #3

(Remarks to the Author)

The authors have addressed all my concerns. I have no further comments and endorse this for publication.

Version 2:

Reviewer comments:

Reviewer #1

(Remarks to the Author)

The manuscripts now is accept for publication.

“Identification of Druggable Binding Sites and Small Molecules as Modulators of TMC1”

We thank the Editorial team and each of the Reviewers for their thorough reading of the manuscript and detailed, considered comments. Below, we provide our point-by-point responses to each specific comment.

Reviewer #1 (Remarks to the Author):

Point 1: The molecular dynamics (MD) simulations were conducted for a duration of 25 ns. This simulated time is insufficient to ascertain the stability of the protein. The authors are advised to extend the simulation period to a minimum of 100 ns to provide a more robust confirmation of protein stability.

We appreciate the reviewer’s suggestion to extend our molecular dynamics (MD) simulations from 25 ns to 100 ns to more robustly assess the stability of the MET complex. In response, we extended the MD simulation to 100 ns and conducted a detailed analysis to verify structural stability.

Our additional 100 ns simulation showed that the MET complex exhibited no significant structural deviations when compared to the 25 ns simulation. The Root Mean Squared Deviation (RMSD) for the backbone atoms, along with the Root Mean Squared Fluctuation (RMSF) and the Radius of Gyration (Rg), remained consistent between the 25 ns and 100 ns timeframes. This stability indicates that the protein reached an equilibrium state during the initial 25 ns simulation and maintained this stability throughout the extended period.

In our revised **Supplementary Figure 2**, we provide a detailed comparison of RMSD, RMSF, and Rg values for all key protein subunits between the 25 ns and 100 ns simulations. Notably, RMSD values for the TMC1 and CIB2 subunits remained under 2 Å, indicating that the core structure of the protein complex is stable. As expected, we observed minor fluctuations in the TMIE subunit due to its single-transmembrane nature and flexible coil regions. However, these fluctuations are within the range reported in the literature for membrane proteins (<https://doi.org/10.1016/j.jmgl.2005.05.006>).

Supplementary Figure 2 [abbreviated]. Molecular dynamics simulations of MET complex. ... (D) Time dependence of the Root Mean Squared Deviation (RMSD) for TMC1, TMIE and CIB2 backbones during the 100ns MDs. **(E)** Radius of

gyration (R_g) to study the compactness of the protein. **(F)** Root Mean Squared Fluctuation (RMSF) characterizing the internal residue fluctuation of TMC1, TMIE and CIB2 during the 100 ns MDs.

Next, we compared the conformations at MD-25 ns and MD-100 ns and detected no major conformational changes (**Figure R1**, not included in the manuscript).

Figure R1. Structural comparison over time of the Initial system used in our study (25 ns with energy constraint of $5 \text{ kcal} \times \text{mol}^{-1} \times \text{\AA}^{-2}$ applied to the secondary structure), the extended MD to reply to reviewers' comments (100 ns with energy constraint of $5 \text{ kcal} \times \text{mol}^{-1} \times \text{\AA}^{-2}$ applied to the secondary structure).

Additionally, we compared the pore dimensions of the TMC1 subunit using HOLE analysis at both 25 ns and 100 ns. Our results confirmed that the pore remained open and stable in both cases, with no significant alterations in pore dimensions or profile (**Supplementary Figure 2**, panels **G-I**). These findings further reinforce that the 25 ns simulation is sufficient to capture the relevant conformational state for our study.

Supplementary Figure 2 [abbreviated]. Molecular dynamics simulations of MET complex. ... TMC1 pore of chain A represented by a blue funnel obtained by HOLE after 25 ns (**G**) and 100 ns (**H**). **(I)** Pore profile of the TMC1 chain A obtained by HOLE analysis. Top, middle, and bottom areas of the pore after 25 ns and 100 ns are labeled.

We acknowledge that longer MD simulations may be necessary for studies focusing on dynamic processes such as ion conductance or ligand-induced conformational changes. However, the purpose of our study was to obtain a stable, open-pore conformation of the MET complex for docking studies. Given that the 25 ns simulation achieved this objective, we believe re-running all docking and experimental validations using the 100 ns frame is unnecessary.

Finally, we want to emphasize the practical implications of our findings. Our study successfully identified and experimentally validated novel channel blockers using a combination of 25 ns MD simulations and *ex vivo* assays. We believe that raising the computational requirements unnecessarily for this study would limit the accessibility of our approach to other researchers, especially those with restricted computational resources.

In summary, based on the extended 100 ns simulation, we conclude that the 25 ns MD simulation duration is adequate for the scope of this study. We have incorporated these additional findings into the revised manuscript (see revised text on page 4, lines 177-185 and updated **Supplementary Figure 2**). We hope the reviewer finds this response satisfactory and thank them once again for their valuable feedback.

Point 2: The authors need to supply references supporting the statements made, as well as the criteria for assessing MD results in the Methods section. Specifically, references are required for the following assertions:

We thank the reviewer for pointing out the need for additional references and clarification regarding the criteria for assessing MD results. We have revised the manuscript to address these concerns and have added appropriate references as follows:

Line 161-163: “When analyzing hetero-subunits independently, both TMC1 and CIB2 protomers remained stable during the 25 ns MDs, with RMSD values at $\sim 2\text{\AA}$. However, fluctuations were observed in the C-terminal TMIE domain, with RMSD values within 4\AA along the trajectory.”

New references #90-92 (line 162-164) were introduced as follows:

“When analyzing hetero-subunits independently, no major changes were evidenced along the 25 ns trajectory, which is in agreement with the literature for MDs of membrane proteins⁹⁰⁻⁹². Both TMC1 and CIB2 protomers remained stable during the 25 ns MDs, with RMSD values at $\sim 2\text{\AA}$.”

Line 166-167: “Root Mean Square Fluctuation (RMSF) analysis, which assesses local atomic fluctuations and flexibility, revealed that TMC1 chains exhibited fluctuations under 2\AA .”

In lines 168-170, the text was updated with a new reference #93 as follows:

“Root Mean Square Fluctuation (RMSF) analysis, which assesses local atomic fluctuations and flexibility, revealed that TMC1 chains exhibited fluctuations under 2\AA similar to its paralog TMEM16A⁹³ (Supp Figure 2C).”

Line 173-175: “The distribution of atoms within the MET complex was assessed using the radius of gyration measurements (Supp Figure 1C), and no major changes were detected, with the radius of gyration remaining within $1\text{\AA}, \dots$ ”.

We complemented and edited the text (lines 177-179) adding new references #90-92, and #94, as follows:

“The spatial distribution of atoms within the MET complex was assessed using the radius of gyration (R_g) measurements (Supp Figure 2B), and no major changes were detected, with R_g values remaining within 1\AA as reported for long 100 ns simulations of other membrane proteins⁹⁴.”

In addition, we included the following statement (line 179-180):

“Our results are in line with those reported for MD simulations of membrane proteins^{90-92,94}.”

We have not identified other references describing both the RMSF and radius of gyration for MD simulations of TMC1. However, our original submission referenced studies #88-89, which provide additional data about the simulations and stability of the MET complex (lines 147-149):

“MmCIB2 was included to increase the stability of the MET complex and preserve its influence in phospholipid dynamics and ion permeation^{88,89}.”

Ref #88. Giese, A. P. J. et al. Complexes of vertebrate TMC1/2 and CIB2/3 proteins form hair-cell mechanotransduction cation channels. *eLife* **12**, (2023).

Ref #89. Walujkar, S. et al. In Silico Electrophysiology of Inner-Ear Mechanotransduction Channel TMC1 Models. 2021.09.17.460860 Preprint at <https://doi.org/10.1101/2021.09.17.460860> (2021).

Point 3: The authors state, “The combination of MD simulations and channel pore analysis by HOLE, has facilitated the identification of three primary target regions within the two elongated TMC1 pores” (Lines 186-187). However, the manuscript lacks a detailed analysis of the channel pore by HOLE. The authors should provide comprehensive details regarding this result.

We appreciate the reviewer’s request for more detailed information regarding the HOLE analysis of the TMC1 channel pore. We conducted the HOLE analysis following standard protocols, which are widely accepted and utilized in the field of ion channel research. Nonetheless, we understand the need to provide additional clarity regarding our results.

In our study, we employed HOLE to analyze the two elongated TMC1 pores and identify key target regions suitable for docking studies. The HOLE algorithm, as referenced in the manuscript, enables the identification of pore dimensions and potential bottlenecks within transmembrane domains, which are crucial for understanding ion permeation and drug binding. This approach aligns with previous studies involving HOLE analysis of ion channels, including TMC1 and other channels see Ref #54 (Figure 4 in <https://www.nature.com/articles/s41586-022-05314-8> and other relevant publications such as <https://doi.org/10.1038/ncomms5377>; <https://doi.org/10.1126/science.1261512>). Specifically, we modeled the pore profiles at different time points to assess stability and identify targetable regions.

While the presented HOLE analysis is sufficient for the purposes of this study, we recognize the importance of further investigation into the TMC1 pore dynamics. Ongoing research in our group focuses on long-term MD simulations combined with applied electric fields to study ion conductance properties. This work includes additional profiling using HOLE and principal component analysis (PCA) along simulation trajectories. These efforts will provide deeper insights into pore dynamics and lipid interactions, expanding on the preliminary findings reported here.

Point 4: In the 3D-pharmacophore modeling, only the APRR pharmacophore comprises four features, while the remaining nine pharmacophores were modeled with three features. Additionally, The authors selected the top 10 pharmacophore models based on the PhaseHypoScore but did not validate the reliability of these models with external databases. Validation results should be provided.

Thank you for your valuable feedback. The most relevant validation of our approach is the positive outcome of our screen, as shown by the *ex-vivo* data provided in this study. However, we have now expanded the “Identification of Common Pharmacophores for MET Channel Block” section of our results to include additional validation of all pharmacophore models (Page 5, line 235-250), also inserted below.

“To validate our pharmacophore models, we tested their ability to discriminate between decoys and known MET channel blockers. We evaluated the models’ performance using the area under the curve (AUC) of the corresponding Receptor Operating Characteristic (ROC) Curve. The AUC value ranges from 0 to 1, with 1 indicating ideal performance and 0.5 indicating random behavior. All AUC values for our models ranged from 0.86 to 0.99 (Supp Table 3), demonstrating that our pharmacophore models can accurately classify compounds as active or inactive. Additionally, all the active compounds (known MET channel blockers) were successfully identified by each pharmacophore model.

We further evaluated the performance of each pharmacophore using the Güner-Henry (GH) scoring method. This metric is a reliable indicator, because it incorporates both the percentage ratio of active compounds in the hit list and the percentage yield of active compounds in a database. The number of active and decoy compounds for each pharmacophore, along with the characteristics for GH (eg, Sensitivity, Specificity, enrichment factor (EF), percentage yield of active compounds (Ya), and % yield of actives) are listed in Supp Table 3. Eight out of the ten pharmacophore models have a GH score higher than 0.7 indicating that these models are good and reliable⁹⁸⁻¹⁰¹. Some studies consider a GH score greater than 0.5 to indicate a good model reliability¹⁰², suggesting that pharmacophores ARR-1 (GH score = 0.675) and HRR (GH score = 0.664) are also valid models.”

In addition, we expanded the methods section (Page 16, line 750-755, also provided below), and added a new Supplementary Table 3.

“To validate our pharmacophore models, we use active compounds (known MET channel blockers) and decoys (40 decoys per active) generated with LIDeB¹⁵⁴ (Supp Table 3). To evaluate the performance of each pharmacophore, different key measures were considered, including the area under the ROC curve (AUC) calculated as described¹⁵⁵; the percentage of actives, the percentage yield of active compounds, sensitivity, specificity, enrichment factor and the Güner-Henry (GH) scoring calculated as reported in the literature¹⁰⁰.”

Point 5: The authors should discuss the structure-activity relationship (SAR) of potential compounds derived from their in silico results and the correlation of these results with the pharmacophore model.

We have discussed the structure-activity relationship of the compounds resulting from our *in-silico* studies, highlighting specific chemical moieties and potential interactions in the pore. Some examples of the SAR, which were already discussed, include:

Line 475: Posaconazole is predicted to establish key contacts within the TMC1 pore (Figure 5D).

Line 481: At the bottom site of the pore, the triazinone ring forms a hydrogen bond with N573 and the hydroxyl group interacts with D540 in the ZN3 pocket.

Line 484: Another non-FDA-approved compound, ZINC24739924, docked at the bottom site of the pore, forming close contacts with residues of TM8 (Figure 5E).

Line 491: A similar binding pattern was observed for another novel TMC1 modulator, ZINC58438263, which has a 3-methyl-anisole moiety and binds to the ZN3 site of TMC1 (Supp. Figure 5).

Line 493: Finally, the FDA-approved drug cepharanthine shared similar interactions to its homolog alkaloid tubocurarine within the top, middle, and bottom sites of the TMC1 pore (Figure 5F).

Line 503: However, it is possible that water molecules could facilitate these interactions at the cytoplasmic region of the ZN3 pocket (Figure 5F).

To follow the reviewer’s comments, the above analysis was complemented with the edited text in lines 549-565.

“From Library 1, the compounds ZINC24739924 (56.25 ± 21.29%), ZINC12986242 (46.1 ± 14.28%), ZINC12430014 (60.94 ± 15.64), ZINC58438263 (46.8 ± 18.59%), showed significant reductions of AM1-43 uptake. From Library 2, posaconazole (60.02 ± 19.12%), pyriothioxine (also called pyriothioxin or pyritinol) (53.52 ± 30.18%), and cepharanthine (48.09 ± 20.72%) exhibited the most promising results. Despite being less effective, ZINC06530230, nefazodone, indinavir, lapatinib, and ceforanide still significantly reduced AM1-43 loading. Interestingly, amitraz from Library 2 appeared to have the opposite effect, increasing AM1-43 loading into OHCs (122.4 ± 21.64%; Supp Figure 8).”

Since these chemical moieties present in the selected compounds matched our pharmacophore models, it suggests that a standard SAR analysis focused only on the most structurally diverse ligands (excluding the receptor) according to our Tanimoto-structural analysis should be complemented with docking results and MM-GBSA studies to better understand the complexity of a Drug-TMC1 system, at the atomic level. Here, a small-molecule modulator could interact with high affinity not only with a specific amino acid in the TMC1 pore, but also with phospholipids. Please see lines 464-470.

Currently, we are planning to explore the structural derivatization of the evaluated compounds from each cluster reported in Figure 7 to better identify a SAR for a hit-to-lead optimization.

Point 7: “The goal of your study is to discover TMC1 modulators”. When you conducted the in silico approach, how did you determine whether the potential compounds identified after screening would act as modulators or inhibitors? Please provide more information on this.

Thanks for the comment. We use the term “modulator” when referring to a compound that may either activate or inhibit the activity. Thus, a compound that inhibits the activity of a protein is a modulator of its activity. The TMC1 modulators we identified are predicted to regulate the activity of TMC1 proteins either by inhibiting or activating it.

With the *in-silico* technique we report, other groups might be able to identify compounds that can either enhance or inhibit the function of TMC1, depending on the desired therapeutic outcome, by changing the parameters of the pharmacophores, or developing pharmacophores based on other known active compounds. Since the open conformation of the TMC1 structure we used for docking revealed 3 potential binding sites, further experimental validation is required to differentiate between activators and inhibitors, including the determination of minimum active concentrations, which cannot be

determined by our *in silico* approach. We do not know if the tested compounds will interact with other protein members of the MET channel, but our molecular docking and MM-GBSA predictions suggest that our compounds bind to the TMC1 pore. Subsequently, our *ex-vivo* experiments suggest that the selected hits reduced the FM1-43 dye uptake. However, since we do not show direct evidence of channel inhibition using electrophysiology, we believe it is more appropriate to refer to compounds as “modulators”. In addition, we do not want to restrict readers to the assumption that our approach can only identify inhibitors due to the use of known MET-blockers for the design of different sets of pharmacophores for *in-silico* compound screening in future studies of TMC1 or other channels.

Reviewer #2 (Remarks to the Author):

The authors present a comprehensive study on identifying potential druggable binding sites and small molecules as modulators of TMC1. However, a few minor issues remain to be addressed.

1. The authors used the AM1-43 dye uptake assay to validate the potential TMC1 modulators in cochlear hair cells. However, this assay has some limitations, such as the low sensitivity and the inability to measure the binding affinity of the compounds. More importantly, whether these compounds have non-specific interactions with TMC1 through aggregation is also a concern.

Thank you for pointing this out. As mentioned above in response to the comments provided by the Reviewer #1, the goal of this study is to present a pipeline that allows for identification of novel channel modulators. We predicted the binding affinities *in silico* using MM-GBSA with and without the influence of phospholipids. We agree that electrophysiology experiments in future studies will help to experimentally determine binding affinities, to be compared with our *in-silico* predictions.

Regarding the potential for non-specific interactions or compound aggregation, we acknowledge the importance of this issue. To prevent aggregation and solubility issues for our compounds, we dissolved them in 2% DMSO. In addition, we used sonication to minimize the possibility of any aggregate in solution that might interfere with the dye loading assay. Our imaging results did not indicate presence of any possible aggregates within the tissue during dye loading experiments. Furthermore, since only hair cells express TMC1 within the cochlear tissue, and we did not observe dye loading into any other cell types, we ruled out non-specific interaction. Our compounds are likely to act at the single-channel level, however further experiments are required in future studies to explore this.

2. In Figure 3, the chemical structure of Amsacrine is not correctly drawn.

Thanks for identifying this, the structure was corrected.

3. In Figure 8, I am unsure whether the protonation states of the compounds Pantoprazole and Pyritinol are correctly assigned.

The protonation states were not assigned manually, but were predicted by the Epik module of Maestro, (please see the details within the methods section). The protonation states showed in each figure correspond to the species that reported the best docking results. We double-checked the protonation states of the compounds and are pleased to inform that all of them are reported correctly. For example, best docking and MM-GBSA results for pyritinol were obtained in the absence of positive charges (Figure 7).

4. Page 11, lines 521-525. I am confused by the inconsistent experimental results of lapatinib, pantoprazole, and pyritinol compounds.

Thank you for your careful interrogation of this dye loading data. The variability is most apparent in Supp. Fig 8, in which we comprehensively detailed all cell fluorescence/AM1-43 loading levels, for each experimental session.

Variability in AM dye loading can be explained by multiple factors, including the functional state of MET channels, cell integrity, temperature, and experimental conditions. To ensure minimal variability, all experiments were performed using mice of the same age and using a standardized culture protocol. We carefully optimized and kept constant the concentration of AM1-43, as well as the exposure times to the dye.

Similar variability was observed in the Kenyon et al. 2021 publication identifying MET channel blockers, from which our starter compounds for the *in silico* search were derived. In Figure 2 of Kenyon et al. 2021, while some compounds protected hair cells from aminoglycosides completely and others poorly, many of the compounds showed a bimodal distribution of hair cell survival between trials (see panels Fig 2C, E and H of Kenyon et al. 2021). Therefore, although it is not clear why some compounds have varied blocking potential in repeated trials, we consider this variation an acceptable limitation of this *in vitro* test platform.

To avoid confusion, we replaced the dye loading data between the Supplementary and main Figures, and provided the number of independent experiments (replicates) for each compound in Figure 8. We also included the necessary controls and the results of the statistical analysis. All compounds were freshly prepared before testing for dye uptake ensuring best quality of the stock samples.

More specifically regarding the variability for pantoprazole, lapatinib, and pantoprazole – it is hard to give specific suggestions, but we can speculate. Pantoprazole showed poor modulation activity in our experiment despite the predicted binding affinity. It is mainly used as an antacid compound known to irreversibly bind and block the H⁺/K⁺ ATPase proton pump of the gastric parietal cells. In solution, under slight acidic conditions, the thionyl group of pantoprazole gets protonated allowing the structural change of the molecule to its active form (a derivative of pantoprazole) <https://pubmed.ncbi.nlm.nih.gov/19938880/>. Thus, it is possible that despite the good binding affinity *in silico* predictions, pantoprazole was transformed to its active form in solution. A similar effect could be speculated for the sulfonyl group of lapatinib. As for pyritinol, it might dissociate to its reduced pyridoxine molecules under light exposure (pyritinol is formed by two molecules of vitamin B6 – pyridoxine, which are linked to each other by a disulfide bond <https://link.springer.com/article/10.1007/s40267-019-00623-x>).

Referee # 3 (Remarks to the Author):

The introduction and the majority of the results are really well written. They are logical and easy to follow. In contrast, the latter section seems rushed and requires some edits.

Thank you. The manuscript has undergone major editing, and we hope the latter sections read more clearly now.

From the section on “Validation of novel TMC1 modulators in cochlear hair cells using AM1-43 dye uptake assay” to the end of the discussion there are a number of sections I find confusing and not easy to follow, these are listed below with other queries and typos.

Queries and Typos:

Line 120 “of” not needed.

We have slightly rephrased the sentence but find the “of” to be necessary.

Line 125: “AM1-43, developed 126 as a fixable analogue of FM1-43, functions effectively in live tissue along with FM1-43.” I’m not sure what you mean by along. Do you mean its acts the same was as FM1-43 or have you added it with FM1-43? I’m assuming the former.

Thank you for pointing this out. The sentence now reads: “AM1-43, developed as a fixable analogue of FM1-43, functions in live tissue similar to FM-143, and is often used for both, live and fixed tissue imaging.”. Page 3, line 126-127.

Line 186-190: “three primary target regions within the two elongated TMC1 pores (Figure 2). Along the Z axis, we have assigned these regions to an expanded top area located near the extracellular region, a narrowed middle area within the transmembrane segment, and a more expanded bottom area near the intracellular region (Figure 2A). – By target regions you mean the top middle and bottom sections of the pore? I think this is best represented by Figure 2B right panel.

Thank you, we have now referenced the correct figure (line 197, and line 200).

Line 244-250: It seems that after optimization you get an extra 314 or 939 molecules but after looking at the phase screen

score these were discarded is that correct? Does this always happen if so why bother identifying the extra if they are only going to be discarded.

That is correct. The extra 314 or 939 structures represent additional 3D conformations across the 4187 compounds (ZINC Database) or of the 1789 drugs (FDA database) after optimization with LigPrep and Epik. For example, we processed 4187 structures with LigPrep and Epik, for each compound. LigPrep and Epik expand tautomeric and ionization states (predicting aqueous phase pKa values and protonation states), ring conformations, and stereoisomers consistent with the input information (4187 ligands) to fully capture the relevant 3D molecular states. This process yielded 314 extra conformations of different molecules from the original set of 4187. Similarly, the FDA dataset yielded 939 additional conformations. Next, we used all these conformations for Phase and selected the most optimal molecule representing the most stable 3D conformation and protonation state that best fit/aligns to a corresponding pharmacophore based on its PhaseScreenScore. This approach allows to predict a unique conformation with a best fit to the pharmacophore for each given molecule to represent its bioactive form, similar to the reference blockers used to build each pharmacophore in 3D.

In summary, we perform this process to identify the conformation that would best fit the pharmacophore making sure we do not overlook good candidates, and discard all other confirmations to minimize the number of molecules for molecular docking and MM-GBSA, significantly reducing computational costs. Otherwise, each of these 313 + 938 structures with molecular docking would yield approximately 10 poses per molecule, resulting in 3130 + 9380 extra docked-optimized structures within the pore of TMC1. Ultimately, only one pose would be selected using the best Glide and Emodel scores (Figure 7) to check the interactions with the amino acids in the top, middle, and bottom area of the TMC1 pore.

The selection of optimized 3D-structures after LigPrep, Epik, docking and MM-GBSA for Tubocurarine is a good example (see Supplementary Table 7). Only the *R* isomer was considered, while both the *R* and *S* isomers for phenoxybenzamine were considered for the analysis, since all commercial preparations of phenoxybenzamine contain the drug as racemate (*R* and *S* isomers).

Line 295: “were” unnecessary.

Corrected to “*This set included 13 blockers ...*”. Page 7, line 326.

Figure 5 E not I for Benzamil.

Corrected.

Line 358: Cepharanthine is not one of the compounds listed as one of the 16 being tested in this section. Where has this come from? It needs explaining.

The reviewer is correct. The section “Predicting binding sites of known MET blockers within the pore region of TMC1”, refers to the analysis of the docking poses of 13 known and chemically distinct MET blockers (from Figure 1) used as a training set for building 3D-pharmacophore models. To increase the diversity of the chemical space, we then included benzamil, tubocurarine, and DHS (see lines 326-328). Cepharanthine belongs to the group of modulators identified following Phase, docking, MM-GBSA, and experimental evaluation. We rearranged the text and the figures to clarify this and moved the HOLE description of the docking poses for cepharanthine to the “Binding modes and affinities of newly identified hit compounds within the TMC1 pore” section, adding “*Similar to FM1-43, molecular docking and HOLE showed that its positively charged nitrogen is located within the predicted pore radius of TMC1 (Supp Figure 6).*” (Page 11 lines 494-496).

Line 406: You state “ligands” it becomes clear later this is the 16 known MET blockers you used above and not the ones you’ve predicted but can you make that clear at the beginning.

We have changed “ligands” to “ligands (the known 16 blockers, listed in Supp Table 7)” (Page 10, line 436).

Line 408: Typo “theses”?

Corrected.

Line 425: Typo space between “Library Iwith”

Corrected.

Line 442: “(Table 4)” for some reason not in bold here.

Corrected. Now Table 2 (line 469).

Line 447: “Our results are provided in the same order as presented on Figure 6.” – confusing can you just say our results are presented in Figure 6 with the known blockers?

We removed the text “Our results are provided in the same order as presented on Figure 6.” And simply cited the Figure 2B and Figure 5, together. (Line 473)

Line 489 – “(M407, S408, G411, P415, T416, I440, L444, N447, L524, T532, T535, G572, N573, A576, F579, M583, and R601) with contact frequencies exceeding 0.5, suggesting they are likely to be among the key residues involved in modulating the TMC1-ligand interaction.” – are there any deafness mutations in these residues? I sort of assume not given you say they are novel.

The reviewer raises a valid point. The residues listed in the sentence refer to novel residues predicted to modulate TMC1-ligand interaction, based on our docking and MM-GBSA results. However, upon reviewing the Clinvar Database, we found that two residues from our list: G417R and R604X in *HsTMC1* (corresponding to G411R and R601X in *MmTMC1*) are associated with hearing loss.

Amongst mouse studies, T416K mutation, is a known variant to cause hearing loss in mice (Ref #86 in our manuscript), as reported by the Fettiplace group (see <https://www.ncbi.nlm.nih.gov/pmc/articles/PMC8152607/>).

We hope this clarifies that while most of the residues we predict to modulate ligand binding are novel, some are already associated with hearing loss according to current databases.

Line 499-500: Where in this experiment has FMI-43 been used? Whether this matters depends on the answer to the question I posed above regarding the nature of AMI-43 compared to FMI-43

We changed it to “... a fluorescent dye loading assay using the FMI-43 analogue, AMI-43 ...”. Page 11, Lines 525-526.

Line 505-510: This is should probably be in the methods section.

Thanks for the suggestion, the text was moved to the methods section.

Line 511-515: This is largely repeating the methods.

Thank you, removed.

Line 507: Why have you used 2% DMSO? As this isn’t a protection experiment and you’ve also used 2% DMSO in the control its probably fine but there is data to suggest that high levels of DMSO can be damaging in these types of assay (doi.org/10.1371/journal.pone.0055359, <https://doi.org/10.1016/j.heares.2007.12.002>). In most of these published assays they use concentrations under 1% or even 0.5% if possible.

The referenced article above reported hour to day-long exposure to DMSO. We used 2% DMSO for 120 seconds total exposure. No hair-cell damage was evidenced in our images. 2% DMSO was used to increase the solubility of our organic compounds, since some showed decreased solubility in aqueous buffers.

Figure 8B: This is quite a confusing and misleading figure. If I understand this correctly each circle represents a cell from one culture and the different colour bars are different independent experiments/cultures. So while there is $n = 3$ for Cepharanthine there is only $n = 1$ for Suvorexant, Pantoprazole, ZINC06530230, ZINC12986242, Amitrazm, Cerforanide, Posaconazole and ZINC12430014. While plotting individual cells keeps the variability for the test compounds averaging the control cells to produce the percentage values loses the variability in the controls which makes this a little misleading. The cells in this case are essentially technical replicates. I'd suggest either you change the analysis to take into account this and therefore keep the variability in both the controls and the experimental (e.g raw values and controls also plotted as one of the bars on your graph followed by a stats model that takes into account the variability in the cells and the variability between cultures/animals for your analysis) or you average the compounds as well. The latter though will mean you only have an $n = 1$ for some of the compounds. You could increase this with more experiments, or can you prove your point without these compounds?

Thank you for your valuable feedback. We appreciate your help in clarifying the figure.

To improve clarity, we have decided to swap the positions of this Figure and the **Supp Figure 8**. The main **Figure 8** now presents the average values for each compound, while our **Supp Figure 8** shows the full variability of the data across all conditions and experimental runs.

Figure 8 displays the averaged, normalized fluorescence of each compound across different experimental days (technical replicates, where $n > 1$). The Kruskal-Wallis test was used to compare these values to the averaged, normalized fluorescence of the controls from the same experimental session. For instance, Lapatinib ($n = 2$, from experiments #3 and #5) is compared only to the control values from the same day. We have also included the number of technical replicates and the total number of OHCs per compound in the histogram bars. Since **Figure 8** does not display the full variability of the compounds and controls, we have also provided **Supplementary Figure 8**, which includes all values for all compounds and controls for each OHC.

The reviewer is correct in noting that we only have $n = 1$ experiment for some compounds, and we agree that additional experiments, including more technical replicates, would further strengthen our findings. However, the primary goal of our study was to validate a screening approach that allows to narrow down from millions of compounds to a select few that can be tested *in vivo*. The combination of our *in vivo* data with our *in-silico* screening results provides strong evidence supporting the effectiveness of our approach. While further studies will be necessary to identify and isolate the best candidate blockers, we believe that the current dataset—despite its limitations—sufficiently validates our approach, as 7 out of 15 compounds tested *in-vivo* significantly modulate the dye uptake.

Line 520: Why do the Library 2 compounds have mean and SD numbers but the Library 1 compounds don't?

We have added the Mean and SD values.

Line 522: Lapatinib has a mean of 66.74 with a SD of $\pm 21.65\%$ this is stretching your 60% cut-off threshold.

We have now removed Lapatinib from the list of “effective” modulators (Please see lines 551-555), as indicated as follows:

*“From Library 2, posaconazole ($60.02 \pm 19.12\%$), pyrithioxine (also called pyrithioxin or pyritinol) ($53.52 \pm 30.18\%$), and cepharanthine ($48.09 \pm 20.72\%$) exhibited the most promising results. Despite being less effective, ZINC06530230, nefazodone, indinavir, lapatinib, and ceforanide still significantly reduced AM1-43 loading. Interestingly, amitraz from Library 2 appeared to have the opposite effect, increasing AM1-43 loading into OHC ($122.4 \pm 21.64\%$; **Supp Figure 8**).”*

Line 527-528: I assume the numbers on the end of the sentence are as follows Posaconazole $39.98 \pm 19.12\%$, Cepharanthine $54.63 \pm 21.45\%$ and ZINC24739924 $47.07 \pm 23.98\%$. The reason I ask is because it conflicts with what's written in line 521-522 where Posaconazole is $60.02 \pm 19.12\%$ and Cepharanthine is $48.09 \pm 20.72\%$. Also I don't think 526-529 adds to what's written above.

The previous section expressed the data in terms of percentage of inhibition, which led to some confusion. To enhance clarity, we have revised the section and carefully verified all values (Please see lines (549-555)).

“From Library 1, the compounds ZINC24739924 ($56.25 \pm 21.29\%$), ZINC12986242 ($46.1 \pm 14.28\%$), ZINC12430014 (60.94 ± 15.64), ZINC58438263 ($46.8 \pm 18.59\%$), showed significant reductions of AM1-43 uptake. From Library 2,

posaconazole ($60.02 \pm 19.12\%$), pyriothioxine (also called pyriothioxin or pyritinol) ($53.52 \pm 30.18\%$), and cepharanthine ($48.09 \pm 20.72\%$) exhibited the most promising results. Despite being less effective, ZINC06530230, nefazodone, indinavir, lapatinib, and ceforanide still significantly reduced AMI-43 loading. Interestingly, amitraz from Library 2 appeared to have the opposite effect, increasing AMI-43 loading into OHCs ($122.4 \pm 21.64\%$; Supp Figure 8.”

Line 529: I think you should have tested some compounds you didn't expect to block based on your model to see if they did as predicted not block.

We have used our computational pipeline to lower the burden for the experimental work, reducing the number of compounds to be tested in animals. However, we have now included a list of decoys as part of our *in silico* validation pipeline (see the Results and Methods sections). This additional step showed that our pharmacophore models are statistically reliable.

The following text were introduced as follows:

- Lines 235-250

“To validate our pharmacophore models, we tested their ability to discriminate between decoys and known MET channel blockers. We evaluated the models' performance using the area under the curve (AUC) of the corresponding Receptor Operating Characteristic (ROC) Curve. The AUC value ranges from 0 to 1, with 1 indicating ideal performance and 0.5 indicating random behavior. All AUC values for our models ranged from 0.86 to 0.99 (Supp Table 3), demonstrating that our pharmacophore models can accurately classify compounds as active or inactive. Additionally, all the active compounds (known MET channel blockers) were successfully identified by each pharmacophore model.

We further evaluated the performance of each pharmacophore using the Güner-Henry (GH) scoring method. This metric is a reliable indicator, because it incorporates both the percentage ratio of active compounds in the hit list and the percentage yield of active compounds in a database. The number of active and decoy compounds for each pharmacophore, along with the characteristics for GH (eg, Sensitivity, Specificity, enrichment factor (EF), percentage yield of active compounds (Ya), and % yield of actives) are listed in Supp Table 3. Eight out of the ten pharmacophore models have a GH score higher than 0.7 indicating that these models are good and reliable⁹⁸⁻¹⁰¹. Some studies consider a GH score greater than 0.5 to indicate a good model reliability¹⁰², suggesting that pharmacophores ARR-1 (GH score = 0.675) and HRR (GH score = 0.664) are also valid models.”

- lines 750-755.

“To validate our pharmacophore models, we use active compounds (known MET channel blockers) and decoys (40 decoys per active) generated with LIDeB¹⁵⁴ (Supp Table 3). To evaluate the performance of each pharmacophore, different key measures were considered, including the area under the ROC curve (AUC) calculated as described¹⁵⁵; the percentage of actives, the percentage yield of active compounds, sensitivity, specificity, enrichment factor and the Güner-Henry (GH) scoring calculated as reported in the literature¹⁰⁰.”

Line 543-549: Other than theaflavin and Ani9 the other TMEM16A modulators showed no similar pharmacophores? If so can you please add a line to this effect for clarity.

Thank you for the comment. As indicated in the manuscript, only theaflavin and Ani9 shared several pharmacophores for TMC1 (similar to cepharanthine). It is important to highlight that all the 10 known modulators of TMEM16A match with at least one of the pharmacophores. We decided to include only the analysis of theaflavin because it matches with 9 of the 10 pharmacophores, while Ani9 matches with 6 of them (the second molecule with the highest Total Phase Screen Score). This has now been clarified on page 12, lines 575-576, and page 13, lines 577-579.

“Our results indicate that both cepharanthine and theaflavin share the same APRR pharmacophore (theaflavin matches with 9 pharmacophores). Although theaflavin lacks a protonatable amine, Epik predicted a protonated carbonyl group instead (Supp Figures 9A-D). Additionally, the ARR-2 pharmacophore exhibited similar vector features between cepharanthine and Ani9, a known inhibitor of TMEM16A (Ani9 matches with 6 pharmacophores) (Supp Figures 9E-H).”

It is, however, possible that more TMEM16A modulators (not evaluated in our study) might share similar pharmacophoric properties to TMC1 modulators.

Line 592: For readability maybe change to “Seven out of 15 hit compounds experimentally tested showed significant reduction of AMI-43 dye loading into hair cells (Supp. Figure 5)”

Done. This is now Supp Figure 8. Please see lines 622-624:

“Seven of the 15 experimentally tested compounds showed a significant reduction in AM1-43 dye loading into hair cells (Supp Figure 8), validating our approach.”

Line 594-596: I find this sentence a little clunky can you reword please.

Done, please see lines 625-627.

“Our in silico findings demonstrate that TMC1 possesses an enlarged cavity with druggable ligand-binding sites, capable of accommodating small molecules^{54,135}. We estimate that the pore’s narrowest dimension is approximately 4.5 Å diameter, which is larger than previously reported pore sizes (Figure 2B).”

Line 604-607: I find this sentence confusing and it doesn’t really follow from the section before can you please reword.

Done. Please see lines 637-641.

“Based on the docking poses of FM1-43 and AM1-43 (Figure 5A, and Supp Figure 5), we infer that the tight packing of both the triethylammonium and the dibutylamine groups at the top binding site of the TMC1 pore, along with the presence phospholipids molecules, may explain why bulkier molecules such as FM3-25 fail to block MET currents⁵⁷. Additionally, this tight packing could contribute to the slower permeation of these bulkier molecules compared to the fast uptake of FM1-43 in hair cells⁵⁸.”

Line 620-624: I find this section confusing, did these point mutations result in deaf mice? What do you mean by site directed mutagenesis in this case are you planning on producing mutant mice with specific mutations? I don’t know how this differs from the point mutation mice mentioned. This needs more of an explanation.

We reworded this section. Please see lines 651-655.

“Our docking results accurately predicted several residues that interact with known MET blockers, which have been well experimentally characterized in mice through point mutations in TMC1^{18,89,114,118,123–126}. These mutations have demonstrated the importance of these residues in permeation and block. Future studies involving site-direct mutagenesis and single-cell electrophysiology will allow to evaluate and further validate the novel contact residues identified in this study.”

Line 653: I don’t understand the “Thus” as this is a bit of a jump from the previous section. This seems a bit tacked on as a paragraph.

We now provide a transition sentence. Line 686.

“TMCs belong to a larger superfamily that includes TMEM16 and TMEM63/OSCA proteins⁶¹”

I find the in silico model very exciting with the potential to reduce the number of animals used in this type of research and also save time and money. I would therefore be interested to know if this model will be available to researchers in the field either as an assessable programme they could run on their compounds or if the authors invite researchers to submit queries to them. Either way it would be nice to have this mentioned in the paper.

Thanks so much for this suggestion. The.pdb coordinate files, pharmacophores and the structure of the ready to use MET channel complex we built and simulated will be available for download from the Ramirez Lab Github repository. We have introduced a “Data Availability” section in our manuscript comply with the Nature Policies.

“Data Availability

The PDB file containing the coordinates of the simulated MET channel complex, and the 3D-pharmacophores used in this study are available for download from the Ramirez Lab Github repository: <https://github.com/ramirezlab/Drug-design-targeting-TMC1>.”

Response to Reviewers
COMMSBIO- 24-2865A

“Identification of Druggable Binding Sites and Small Molecules as Modulators of TMC1”

We thank the Editorial team and each of the Reviewers for their responses. We also appreciate the positive comments from all Reviewers and are pleased to learn that the changes incorporated in the previous version of the manuscript addressed all concerns raised by Reviewers #2 and #3 to their satisfaction.

Below, we provide our point-by-point responses to Reviewer #1.

Reviewer #1 (Remarks to the Author):

Thank you for addressing my comments and providing detailed responses. Your efforts to clarify and improve these aspects of the manuscript are greatly appreciated.

Point 1: I appreciate your efforts to extend the MD simulations to 100 ns and to demonstrate stability between 25 ns and 100 ns. However, it is necessary to include the results of the 100 ns MD simulations in the manuscript for the following reasons:

1. As you mentioned: “Next, we compared the conformations at MD-25 ns and MD-100 ns and detected no major conformational changes (Figure R1, not included in the manuscript).”

The figure only illustrates the protein structures at selected time frames (0-12-24 ns for MD-25 ns and 0-25-50-100 ns for MD-100 ns). What evaluation metrics led to the conclusion of “no major conformational changes”? Why did you choose the conformations at 0-12-24 ns for MD-25 ns and 0-25-50-100 ns for MD-100 ns? If the goal is to compare conformations throughout the simulation time, it would be more appropriate to identify time frames showing significant changes and to compare these conformations using specific quantitative evaluation.

Thank you for your comments. We agree and have included the abovementioned Figure R1 into the manuscript as part of the Supplementary Figure 2 (panels I and J), as recommended. In addition, we introduced an additional panel H, to provide a quantitative analysis of the TMC1 pore during the entire simulation period. These data demonstrate the stability of the TMC1 protomers forming the pore, showing no major conformational changes across the 25 ns and 100 ns MD simulations. This was done by calculating the pore radius every nanosecond and plotting the average pore radius with the standard deviation for both pores of TMC1 chain A. In our previous submission, we showed that no structural changes were evidenced in both pores of TMC1 chain A and B at a single time-frame level (Figure 2B). We also included the RMSD, Rg, and RMSF plots (for each amino acid) for TMC1, CIB2, and TMIE for both 25 ns and 100 ns MD simulations.

In panel H, the top, middle, and bottom sites are color-coded as in Figure 4B. For consistency, we added the color-coding and residue labels in Figure 2B, and relabeled the “top, middle, and bottom areas” to “top, middle, and bottom sites” across the manuscript (lines 209-212).

The figure legend was updated to include the figure legends for added panels, as indicated below:

“...(G) TMC1 pore of chain A represented by a blue funnel obtained by HOLE after 25 ns and 100 ns. (H) Pore radius measurements (average \pm standard deviation) of the TMC1 chain A obtained by HOLE analysis over 25 ns and 100 ns simulation trajectories. Measurements were obtained every nanosecond throughout the simulations. The top, middle, and bottom sites of the pore are color-coded as in Figure 2B and Figure 4. The zero point (0 Å) on the z-axis represents the reference position of the middle site of the TMC1 pore at the center of the plasma membrane. (I-J) General view and structural comparison of three representative frames from the 25 ns and 100 ns MD simulations, respectively, revealing no major structural changes between the simulations time points and suggesting stability across the trajectories.”

Supplementary Figure 2 (abbreviated).

We expanded the *Molecular dynamics simulations* section in the Methods (highlighted in yellow) and further provide more details in lines 190-196 as follows:

“This stability can be attributed, in part, to the constraints applied during the simulation, ensuring the stability of the ion channel, and resulting in minimal, if any, conformational differences within the TMC1 protomers and its pores during 25 ns vs 100 ns simulation periods (**Supp Figure 2G-J**). Notably, our *HOLE*⁹⁷ analysis demonstrated that the calculated average pore radius (\pm standard deviation) remained consistent, with no major changes in the pore size distribution. This was assessed at 1 nanosecond intervals for the pore of chain A, and compared across the 25 ns and 100 ns trajectories (**Supp Figure 2H**).”

Please also see lines 755-758 (Methods):

“Pore radius measurements of the TMC1 chain A were obtained over 25 ns and 100 ns production trajectories to determine the stability of the pore and the TMC1 dimer. Measurements were obtained every nanosecond throughout the simulations.”

2. Regarding the RMSD values, it is evident that after 40 ns, the backbone stability was achieved for most proteins, except for TIMEC(A), which did not stabilize during the 100 ns simulation. Specifically, TIMED(B) and CIB2F(B) exhibited significant fluctuations in the initial 40 ns, ranging from 1 Å to 2.5 Å and 2.0 Å, respectively, before stabilizing at these values afterward. These observations indicate that stability of the backbone was generally reached after 40 ns, not 25 ns. Moreover, TIMEC(A) displayed persistent fluctuations and failed to stabilize throughout the entire simulation.

These findings highlight the importance of extending the simulation time to 100 ns and necessitate including this result in the manuscript to provide a comprehensive analysis. The other points are satisfactory.

We agree that adding the 100 ns simulation results improved the manuscript and will help the readers to design their future studies in accordance with their needs.

We agree that both TMIE and CIB2 subunits undergo some conformational changes throughout our simulations, but these are typical of the system. Although TMIE and CIB2 are part of the MET complex with TMC1, we want to clarify that our drug screen was carried out within the top, middle, and bottom sites of the stable TMC1 pore (across the 25 and 100 ns trajectories), and is not affected by RMSF fluctuations of TMIE and CIB2. Both TMIE and CIB2 are further away from the abovementioned TMC1 druggable pore, the binding sites, and the gridbox we used for the screen.

We acknowledge the structural movements mentioned by the Reviewer. However, we want to stress that these movements are not a synonym of an unstable structure, but rather a flexible conformation of TMIE, which we report in our work, and likely due to short remodeling of CIB2 over the trajectories, as explained below.

The two single-pass TMIE subunits (Figure 2) reside on the periphery of the complex and span the membrane, nearly “floating” on the periphery of the complex, flanking each TMC1 subunit, as reported in reference #55 and shown in our Figure 2 (in orange). The majority of TMIE segment is situated outside the gridbox employed for the *in-silico* screening (See Figure 2C-E). Thus, while small fluctuations of TMIE are expected due to its flexible cytosolic helix (RMSD within 4 Å), doubling the RMSD of the tighter CIB2+TMC1 complex (RMSD ~ 2 Å), we believe these fluctuations do not affect the pore, or our docking results.

In lines 174-175, the text was rephrased to:

“This behavior was likely attributable to the “elbow-like” linker⁵⁵ as a new flexible component of the MET complex, allowing for the TMIE cytoplasmic helix to move more freely (Figure 2C, and Supp Figure 2A).”

This is also further discussed in lines 180-184:

“In contrast, the C-terminal cytoplasmic helix of TMIE displayed larger fluctuations, likely because it is positioned outside the bilayer membrane and is exposed to solvent, hinging about a flexible region (Figure 2C and Supp Figure 2C). Similarly, the CIB2 protomers displayed comparable RMSF fluctuations in their N-terminal domains, likely due to the random-coil configuration of these regions (Supp Figure 2C).”

Furthermore, our results agree with recent Nuclear Magnetic Resonance (NMR) experiments and simulations showing extensive remodeling of the CIB2 structure with TMC1 (reference #21). Although CIB2 is important in hearing function and complex formation with TMC1, 100 ns long simulations with pure POPC bilayers showed permeation of K⁺ in either one or both pores of the TMC1 dimer, with or without CIB2 (references #21 and #90, respectively). Thus, while fluctuations in CIB2 during the first ~40 ns are possible and valid, they do not suggest an unstable complex.

Importantly, the flexibility of TMIE shown in our simulations is of great value compared to the 100-ns-long MD simulations of the TMC1+CIB2 complex or TMC1 alone reported in references #21 and #90 that do not include TMIE. In our manuscript, we show new and unique structural insights of the mammalian MET complex by including the TMIE components in our simulations, analyzing the implications of the “elbow-like” linker of TMIE reported in reference #55.

We have included these considerations (highlighted in yellow) in lines 35-41 (abstract), 159-161 (results section), and lines 675-685 (discussion section), respectively.

To expand the TMIE analysis we rephrased (highlighted in yellow) some paragraphs in the following lines 57-60; lines 136-141; lines 342-343; and lines 822-823.

Two new references #138 and #139 were added, and Reference #143 was updated from BioRxiv to PNAS.